# Reaction hijacking inhibition of *Plasmodium falciparum* asparagine tRNA synthetase

Malaria poses an enormous threat to human health. With ever increasing resistance to currently deployed drugs, breakthrough compounds with novel mechanisms of action are urgently needed. Here, we explore pyrimidine-based sulfonamides as a new low molecular weight inhibitor class with drug-like physical parameters and a synthetically accessible scaffold. We show that the exemplar, OSM-S-106, has potent activity against parasite cultures, low mammalian cell toxicity and low propensity for resistance development. In vitro evolution of resistance using a slow ramp-up approach pointed to the *Plasmodium falciparum* cytoplasmic asparaginyl-tRNA synthetase (*Pf*AsnRS) as the target, consistent with our finding that OSM-S-106 inhibits protein translation and activates the amino acid starvation response. Targeted mass spectrometry confirms that OSM-S-106 is a pro-inhibitor and that inhibition of *Pf*AsnRS occurs via enzyme-mediated production of an Asn-OSM-S-106 adduct. Human AsnRS is much less susceptible to this reaction hijacking mechanism. X-ray crystallographic studies of human AsnRS in complex with inhibitor adducts and docking of pro-inhibitors into a model of Asn-tRNA-bound *Pf*AsnRS provide insights into the structure-activity relationship and the selectivity mechanism.

Malaria is a devastating disease. In 2021, *Plasmodium falciparum*, the most deadly of the malaria species, affected more than 200 million people and caused more than 600,000 deaths, mostly of African children[1]. Disruptions to funding and services due to the COVID-19 pandemic exacerbated the problems caused by widespread resistance of parasites to currently used therapies[2], as well as resistance of the mosquito vectors to pyrethroid insecticides[3]. In particular, the recent emergence in Africa of artemisinin resistance-conferring K13 mutations[4,5] is of great concern. There is an urgent need to develop new antimalarial compounds with novel mechanisms of action.

A recent study showed that some *P. falciparum* aminoacyl-tRNA synthetases (aaRSs) are susceptible to reaction hijacking by nucleoside sulfamates[6]. Tight-binding nucleoside sulfamate-amino acid adducts are generated in the active site, thereby blocking enzyme activity. A *Plasmodium*-specific reagent, ML901, was identified that hijacks *P. falciparum* tyrosine-tRNA synthetase (*Pf*TyrRS). By contrast, *Homo sapiens* TyrRS (*Hs*TyrRS) does not catalyze formation of the adduct. X-ray crystallography revealed that differential flexibility of a loop over the catalytic site may underpin differential susceptibility to reaction-hijacking by ML901[6].

Here, we explored a new chemical class of reaction hijacking inhibitors. OSM-S-106 (Fig. 1A) is an aminothieno pyrimidine benzene sulfonamide, with activity against *P. falciparum* cultures. OSM-S-106 was first identified as part of a screen of compounds from a GSK library (originally tagged as TCMDC-135294[7]). While OSM-S-106 is structurally divergent from the nucleoside sulfamates previously shown to target *Pf*aaRSs via the reaction hijacking mechanism, mass spectrometry-based identification of covalent adducts and biochemical analyses revealed that Asn-tRNA-bound *Pf*AsnRS is indeed susceptible to attack by OSM-S-106. By contrast, *Hs*AsnRS is much less susceptible to hijacking by OSM-S-106.

AsnRSs are class II aaRSs, characterized by an α/β fold, with a highly conserved active site. We solved, for the first time, the crystal structure of *Hs*AsnRS in complex with the natural intermediate, Asn-AMP, as well as with synthetically generated Asn-OSM-S-106, providing insights into ligand-induced changes in the enzyme structure. We

✉ e-mail: ewinzeler@health.ucsd.edu; mgriffin@unimelb.edu.au; matthew.todd@ucl.ac.uk; ltilley@unimelb.edu.au

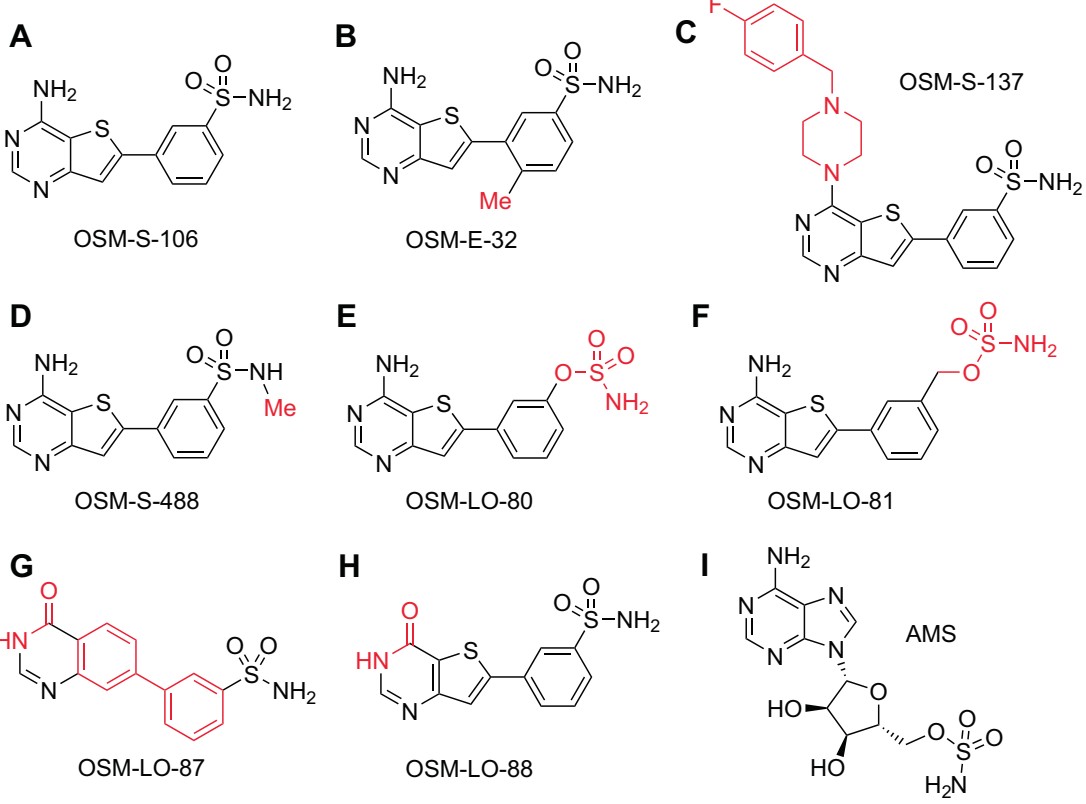

**Fig. 1 | Structures of OSM-S-106, OSM-S-106 derivatives and adenosine 5'-sulfamate. A** OSM-S-106. **B** OSM-E-32. **C** OSM-S-137. **D** OSM-S-488. **E** OSM-LO-80. **F** OSM-LO-81. **G** OSM-LO-87. **H** OSM-LO-88. **I** AMS. Structural differences between OSM-S-106 and derivatives (**B**–**H**) are highlighted in red.

generated an AlphaFold model of the *Pf*AsnRS dimer. *Pf*AsnRS harbors a large insert adjacent to motif I that is predicted to affect the dynamics of binding of substrates and release of products from the active site. This *Plasmodium*-specific structural feature may underpin differences in susceptibility to reaction hijacking. We generated a molecular model of *Pf*AsnRS in complex with the native product, Asn-tRNA; and docked OSM-S-106 and derivatives into the AMP-binding site. This analysis provided insights into the structure-activity relationships (SAR) that underpin the potency of OSM-S-106.

## Results

### Selection of OSM-S-106

A previous screen of 2 million GSK Public Limited Company (PLC) compounds against *P. falciparum* cultures yielded the Tres Cantos Antimalarial Set (TCAMS) library with 13,500 active compounds[7]. One of these compounds, OSM-S-106 (TCMDC-135294; Fig. 1A), was considered attractive from a medicinal chemistry perspective due to its ligand efficient structure[8], its synthetically accessible scaffold and its drug-like properties (Supplementary Table 1). Consequently, OSM-S-106 was chosen as the subject of an Open Source Malaria campaign[8–10].

### Synthesis and characterization of OSM-S-106 and derivatives

Synthesis of OSM-S-106 and its derivatives was achieved using the aminothieno pyrimidine synthesis protocol, as delineated in the Supplementary Material. Briefly, the pyrimidine cores for the OSM-S-106 were prepared using a two-step heterocycle synthesis, followed by bromination and amination and finally a Suzuki coupling with benzenesulfonamide pinacol boronate, to yield OSM-S-106 in yields ranging from 45 to 70%. The synthetic sequence was conducted on a gram scale, giving 200 mg or more of the final product. The molecule appears stable in the solid state with respect to degradation under ambient conditions (Supplementary Material and Supplementary Data 1).

### OSM-S-106 exhibits selective activity against Plasmodium blood and liver stages and low intrinsic clearance by human microsomes

We confirmed that OSM-S-106 exhibits good activity against the 3D7 line of *P. falciparum* (50% inhibitory concentration ($IC_{50\_72h}$) = 0.058 ± 0.017 μM; n = 8). Importantly, OSM-S-106 also prevented the development of *P. berghei* in liver cells (HepG2-A16-CD81-EGF; $IC_{50}$ = 0.25/0.42 μM; n = 2, Table 1). OSM-S-106 exhibited low cytotoxicity against the HepG2 cell line ($IC_{50}$ = 49.6/47.3 μM; n = 2), indicating a selectivity index ($IC_{50}^{HepG2}/IC_{50}^{PbLuc}$) of over 140-fold (Table 1). OSM-S-106 is stable during incubation with human microsomes ($t_{1/2}$ = 395/619 min; n = 2); but shows rapid intrinsic clearance in mouse microsomes ($t_{1/2}$ = 19.7/20.4 min; n = 2) (Supplementary Table 2).

### SAR analysis reveals compound features that are needed for potent activity

Several substitutions of the aminothienopyrimidine core, the pendant aromatic ring and the primary sulfonamide were prepared, as described in the Supplementary Information, to establish SAR, with a readout of activity against 3D7 cultures. The addition of a methyl group to the thiophene ring (OSM-E-32; Fig. 1B), with the aim of reducing compound planarity, was not tolerated (Table 1), nor was the addition of a methyl group to the primary sulfonamide (OSM-S-488; Fig. 1D). A compound bearing a larger amine substituent on the pyrimidine ring (OSM-S-137, Fig. 1C) showed low (though not zero) activity ($IC_{50}$ = 4.4 ± 3.0 μM). Conversion of the sulfonamide to either a sulfamate (OSM-LO-80; Fig. 1E; 5.1 ± 3.2 μM) or a sulfamate with an

**Table 1 | Cytotoxicity of OSM-S-106 and derivatives against *P. falciparum* (3D7) and the HepG2 mammalian cell line**

| Compound | OSM-S-106 | OSM-E-32 | OSM-S-137 | OSM-S-488 | OSM-LO-80 | OSM-LO-81 | OSM-LO-87 | OSM-LO-88 |
|---|---|---|---|---|---|---|---|---|
| *Pf*3D7 IC$_{50(72\,h)}$ (µM) | 0.058 ± 0.017 ($n$ = 8) | 8.3/8.3 ($n$ = 2) | 4.4 ± 3.0 ($n$ = 4) | 12.7/13.8 ($n$ = 2) | 5.1 ± 3.2 ($n$ = 8) | 0.93 ± 0.24 ($n$ = 6) | >25 ($n$ = 6) | 18.7 ± 1.8 ($n$ = 6) |
| HepG2 IC$_{50(72\,h)}$ (µM) | 49.6/47.3 ($n$ = 2) | N/A | N/A | N/A | 17.2/18.2 ($n$ = 2) | 43.4/43.9 ($n$ = 2) | N/A | N/A |
| *P. berghei* liver stage IC$_{50(72\,h)}$ (µM) | 0.25/0.42 ($n$ = 2) | N/A | N/A | N/A | N/A | N/A | N/A | N/A |

$n$ = number of biological repeats. Where available, data are expressed as mean ± SD.

extended linker (OSM-LO-81; Fig. 1F; 0.93 ± 0.24 µM) decreased inhibitory activity compared with OSM-S-106 (Table 1).

An hydroxyquinazolinyl benzene sulfonamide, MMV026546, was also identified in the TCAMS library. Here we resynthesised this pyridone sulfonamide (renamed OSM-LO-87, Fig. 1G); and also generated the corresponding oxo-thienopyrimidinyl benzene sulfonamide (OSM-LO-88, Fig. 1H). OSM-LO-87 exhibited no antimalarial potency, suggesting that the initial report was a false positive; while OSM-LO-88 exhibited very low activity 18.7 ± 1.8 µM (Table 1). Given the marked sensitivity of OSM-S-106 to substitution, we continued the characterization of the initial hit, and sought to identify the target to better understand the requirements for activity.

### OSM-S-106 exhibits a low propensity for developing resistance

In vitro evolution and whole-genome sequencing has been used extensively to explore *P. falciparum*'s propensity for developing resistance and to identify novel antimalarial drug targets and resistance mechanisms[11,12]. A single-step selection was set up, using 10$^7$ Dd2-B2 parasites in each well of a 24-well plate, with OSM-S-106 at a concentration of 3 × IC$_{90}$ (508 nM). No recrudescent parasites were observed over a 60-day selection period.

To validate our protocol, a reference selection was run in parallel using 2 × 10$^5$ Dd2-B2 parasites in each well of a 96-well plate with a *Plasmodium*-specific dihydroorotate dehydrogenase inhibitor (DSM265[13];) at 5 × IC$_{50}$ (58 nM). This yielded 14 recrudescent wells, which corresponds to a Minimum Inoculum for Resistance (MIR) of 1.4 × 10$^6$ parasites required to obtain resistance. Whole-genome sequence analysis provided evidence of amplification events encompassing the dihydroorotate dehydrogenase (dhodh) locus, consistent with increased IC$_{50}$ values for DSM265 (Supplementary Tables 3 and 4 and Supplementary Data 2). Based on these studies, we conclude that the MIR value for OSM-S-106 is > 2.4 × 10$^8$.

### Parasites selected against OSM-S-106 in a slow ramp-up method acquire mutations in asparaginyl-tRNA synthetase (*Pf*AsnRS) and nucleoside transporter 4 (*Pf*NT4)

We next employed a gradual ramp-up exposure method, which has been reported to increase the success of evolving resistant parasites[14]. Starting at the IC$_{50\,72h}$ concentration, we gradually increased to four times the IC$_{50\,72h}$ concentration over 2 months in a Dd2 genetic background (10$^9$ parasites). This ramp-up selection yielded 21 newly emerged coding variants in 10 unique core genes (Table 2; Supplementary Table 5 and Supplementary Data 3 and 4). Sequencing six clones (2 from flask 3 and 4 from flask 2) and comparing Single Nucleotide Variants (SNVs)/Insertions or Deletions (Indels) and Copy Number Variants (CNVs) present in the clones relative to their isogenic parents identified two candidate genes of interest. Parasites from both flasks contained a missense mutation (S22C or H320L) in PF3D7_0103200, which encodes *P. falciparum* nucleoside transporter 4 (*Pf*NT4). In addition, parasites from both flasks contained mutations in the PF3D7_0211800 locus, which encodes *P. falciparum* cytoplasmic asparaginyl-tRNA synthetase (*Pf*AsnRS). All four Dd2-OSM-2 clones harbored an R487S change in

*Pf*AsnRS, while both Dd2-OSM-3 clones and one of the Dd2-OSM-2 clones had a Copy Number Variant (CNV) across a genomic segment on chromosome 2 that contains *Pf*AsnRS. This CNV has never been reported before nor have SNVs in *Pf*AsnRS. Of interest, the precise boundaries of the CNV region varied between clones, suggesting independent events. The likelihood of missense mutations and CNVs in the same gene by chance is extremely low. Based on our experience with hundreds of selections we hypothesized that *Pf*NT4 is more likely a drug resistance gene and that *Pf*AsnRS is the target.

### OSM-S-106 inhibits protein translation and induces the amino acid starvation response

To investigate *Pf*AsnRS as a potential target, we examined the ability of OSM-S-106 to inhibit protein translation. We employed an in-cell assay of protein translation in *P. falciparum* trophozoites, monitored by incorporation of a clickable derivative of the puromycin homolog, O-propargyl-puromycin (OPP)[15,16]. Following a 6 h exposure, protein translation is inhibited with an IC$_{50}$ value of 0.51 µM (Fig. 2A), consistent with *Pf*AsnRS being the target. The same pulsed exposure to OSM-S-106 leads to loss of viability in the next cycle; albeit with a higher IC$_{50}$ value (4.7 µM; Fig. 2A). The data are consistent with previous reports showing that even short-term inhibition of aaRSs is lethal[6,16]. As a control, we showed that pulsed (6 h) exposure to WR99210, an inhibitor of *Pf*DHFR, prevented replication into the next cycle, but had no short-term effect on protein translation (Supplementary Fig. 1A). Exposure to cycloheximide, which inhibits protein translation by interfering with the ribosome, also inhibited OPP incorporation; however, a 6-h exposure had no effect on viability (Supplementary Fig. 1B).

Inhibition of aminoacyl-tRNA synthetases leads to a build-up of uncharged tRNA, which in turn leads to eIF2α phosphorylation[6,17,18]. OSM-S-106 exposure triggers eIF2α phosphorylation, to a similar extent as the known threonyl-tRNA synthetase inhibitor, borrelidin (Fig. 2B and Supplementary Fig. 1C). OSM-S-137, a derivative of OSM-S-106, with lower activity (Fig. 1C), has no effect under the same exposure conditions (Fig. 2B and Supplementary Fig. 1C). Taken together, the data are consistent with OSM-S-106 targeting *Pf*AsnRS.

### OSM-S-106 hijacks the catalytic activity of *P. falciparum* aminoacyl-tRNA synthetases

The identification of *Pf*AsnRS as a potential target of OSM-S-106 was of particular interest to our team given that the compound bears a primary sulfonamide attached to an aromatic ring structure, reminiscent of nucleoside sulfamates, such as ML901 and adenosine 5′-O-sulfamate (AMS), that have been shown to be pro-inhibitors of aaRSs[6] (Fig. 2C). We therefore considered the possibility that OSM-S-106 might exert its activity against *Pf*AsnRS via a reaction hijacking mechanism. Such a mechanism would be expected to generate an Asn-OSM-S-106 conjugate (Fig. 2D). We treated *P. falciparum* cultures with 1 µM or 10 µM OSM-S-106 for 3 h and used targeted mass spectrometry to search for the 20 possible amino acid conjugates. Extracts were subjected to liquid chromatography-coupled with mass spectrometry (LC-MS) and the anticipated masses were interrogated. The extract yielded a strong

**Table 2 | Mutations identified in Dd2 parasites selected with OSM-S-106 and quality metrics for each sequenced parasite line**

| Parasite name | Gene | Gene | Gene | CNV | IC$_{50}$ (µM) | IC$_{50}$ (fold increase) |
|---|---|---|---|---|---|---|
| Dd2-B2-2 | NRS parent | NT4 Parent | GDH Parent | None | 0.079 ± 0.012 ($n$ = 4) | 1.0 |
| Dd2-OSM-2A6 | NRS R487S | NT4 H320L | GDH Parent | None | 0.70 | 8.8 |
| Dd2-OSM-2B2 | NRS R487S | NT4 H320L | GDH Parent | None | 0.70 | 8.8 |
| Dd2-OSM-2A9 | NRS R487S | NT4 Parent | GDH Parent | NRS | 0.32 | 4.1 |
| Dd2-OSM-2D6 | NRS R487S | NT4 H320L | GDH Parent | None | 0.78 | 9.9 |
| Dd2-B2-3 | NRS parent | NT4 Parent | GDH Parent | None | 0.08 | 1.0 |
| Dd2-OSM-3E5 | NRS parent | NT4 S22C | GDH D200Y | NRS | 0.16 | 2 |
| Dd2-OSM-3H7 | NRS parent | NT4 S22C | GDH D200Y | NRS | 0.22 | 2.8 |

Six clones from two independent OSM-S-106-pressured cultures were sequenced and analyzed to identify potential resistance-conferring variants. Variants with ≥90% alleles mapping to the alternate allele are shown. NRS = *Pf*AsnRS; NT4 = *Pf*NT4; GDH - *Pf* glutamate dehydrogenase 3.

signal for Asn-OSM-S-106, with a precursor ion at *m/z* 421.0746, retention time at 4.9 min and fragmentation spectrum consistent with that of the synthetic Asn-OSM-S-106 conjugate (Fig. 2E and Supplementary Fig. 2A). In the samples with 10 µM OSM-S-106 treatment, minor MS peaks were also detected for the adducts of glycine and alanine along with MS/MS spectra containing characteristic OSM-S-106 ion at *m/z* 307 (Supplementary Fig. 2B–E), suggesting that these GlyRS and AlaRS are also weakly susceptible to inhibition via the reaction hijacking mechanism.

### Further characterization of OSM-S-106 targets

To investigate *Pf*AsnRS as a target, transfectants harboring the *Pf*AsnRS[R487S] mutation were generated in a Dd2 parent line (Supplementary Fig. 3A–C). The mutant line exhibited a 2.3-fold decreased sensitivity to OSM-S-106 (Fig. 2F and Supplementary Table 6), consistent with *Pf*AsnRS being an important target.

We used the TetR/DOZI-RNA aptamer module to conditionally regulate the expression of some of the *P. falciparum* gene products in which mutations arose during evolution of resistance (see Table 2), namely *Pf*AsnRS, *Pf*GDH3 and *Pf*NT4. We also modulated the level of cytoplasmic *Pf*AlaRS (PF3D7_1367700) and *Pf*GlyRS (PF3D7_1420400), given that we observed production of a low level of adducts of OSM-S-106 with these amino acids. OSM-S-106 contains a sulfonamide group that is predicted to bind tightly to carbonic anhydrase[19], suggesting *P. falciparum* carbonic anhydrase (*Pf*CA; PF3D7_1140000) as another possible target, so, we also knocked down *Pf*CA. Luminescence-based viability assays revealed that knockdown of the cytoplasmic aaRSs perturbs the growth of parasites (Supplementary Fig. 3D), indicating the genes are essential for blood stage development. By contrast, knockdown of *Pf*NT4, *Pf*GDH3 and *Pf*CA did not have a significant impact on parasite growth, consistent with previous studies for *Pf*NT4 and *Pf*GDH3[20,21].

Upon knockdown of *Pf*AsnRS, the parasites exhibited a 6.6-fold enhancement in susceptibility to OSM-S-106 compared to the control (Fig. 2G and Supplementary Table 7), confirming the inhibitory interaction. *Pf*NT4 knockdown also sensitized the parasites to OSM-S-106 (3-fold shift; Fig. 2H and Supplementary Table 7). Differential sensitivity to OSM-S-106 was not observed following knockdown of *Pf*AlaRS, *Pf*GlyRS, *Pf*GDH3 or *Pf*CA (Supplementary Fig. 3E–H and Supplementary Table 7), arguing against these proteins being important targets of the inhibitor.

### Recombinant *Hs*AsnRS has very limited capacity to generate Asn-OSM-S-106 adducts compared to *Pf*AsnRS

Using an *E. coli* expression system, we generated recombinant *Pf*AsnRS and *Hs*AsnRS. Following removal of the His-tag and generation of wildtype enzymes, analytical ultracentrifugation revealed that the proteins are dimeric in solution (Supplementary Fig. 4). We used targeted mass spectrometry to examine the ability of recombinant

*Pf*AsnRS and *Hs*AsnRS to generate the Asn-OSM-S-106 conjugate. The enzymes were incubated with ATP, Asn and *E. coli* tRNA in the presence of OSM-S-106 (10 µM). Following precipitation of the tRNA and protein, the supernatants were subjected to LC-MS analysis. For *Pf*AsnRS, we detected a peak at *m/z* 421.0735 with retention time at 7.1 min, consistent with that of the Asn-OSM-S-106 standard (Supplementary Fig. 5A). The identity of the adduct was further confirmed by MS/MS analysis compared with the synthetic standard (Supplementary Fig. 5B). By contrast, a signal with 18-fold lower intensity was detected for Asn-OSM-S-106 when *Hs*AsnRS was incubated with OSM-S-106 under the same conditions (Supplementary Fig. 5C).

### OSM-S-106 inhibits ATP consumption by *Pf*AsnRS but not *Hs*AsnRS

We assessed the ability of the recombinant aaRSs to consume ATP in the initial phase of the aminoacylation reaction, *i.e.*, via the formation and release of AMP. For these studies, we used recombinant versions of *Pf*AsnRS and *Pf*AsnRS[R487S], the mutant selected during evolution of resistance to OSM-S-106, as well as *Hs*AsnRS. In the absence of tRNA, *Pf*AsnRS, *Pf*AsnRS[R487S] and *Hs*AsnRS consume low levels of ATP (Fig. 3A). Addition of *E. coli* tRNA substantively increases the level of ATP consumption (Fig. 3A), consistent with productive aminoacylation.

OSM-S-106 inhibits consumption of ATP by wildtype *Pf*AsnRS when added in the presence of tRNA, but not in its absence (Fig. 3B). This is consistent with a reaction hijacking mechanism whereby enzyme-bound amino acid-conjugated tRNA is the target of nucleophilic attack by OSM-S-106 (Fig. 2C). The *Pf*AsnRS[R487S] mutant is inhibited less efficiently than wildtype *Pf*AsnRS (Fig. 3B), consistent with the decreased sensitivity of cultures of *Pf*AsnRS[R487S] mutants to OSM-S-106 (Supplementary Table 6). OSM-LO-80, which exhibits weaker antimalarial potency (Table 1) also shows weaker inhibition of the consumption of ATP by *Pf*AsnRS (Fig. 3C). OSM-S-106 and OSM-LO-80 do not inhibit consumption of ATP by *Hs*AsnRS (Fig. 3C).

We previously showed that AMS (Fig. 1I) is a broadly reactive pro-inhibitor that hijacks a range of aaRSs in both *Plasmodium* and human cell lines[6]. Here we used AMS, generated as previously described[22], and kindly provided by Dr Steven Langston, Takeda Pharmaceuticals, as a positive control for reaction-hijacking inhibition of aaRSs. When added in the presence of all substrates, AMS inhibits ATP consumption by both *Pf*AsnRS and *Hs*AsnRS; although *Pf*AsnRS appears to be more susceptible (Fig. 3D). These data suggest that *Hs*AsnRS is intrinsically less susceptible to reaction hijacking; and that OSM-S-106 has structural features that exploit that difference in susceptibility, providing selectivity.

Synthetically generated Asn-OSM-S-106 strongly inhibits the activity of *Pf*AsnRS, *Pf*AsnRS[R487S] and, to a lesser extent, *Hs*AsnRS (Fig. 3E, F), suggesting that the susceptibility to reaction hijacking depends largely on the ability of the enzyme to generate the Asn-OSM-S-106 adduct, rather than the ability to bind the

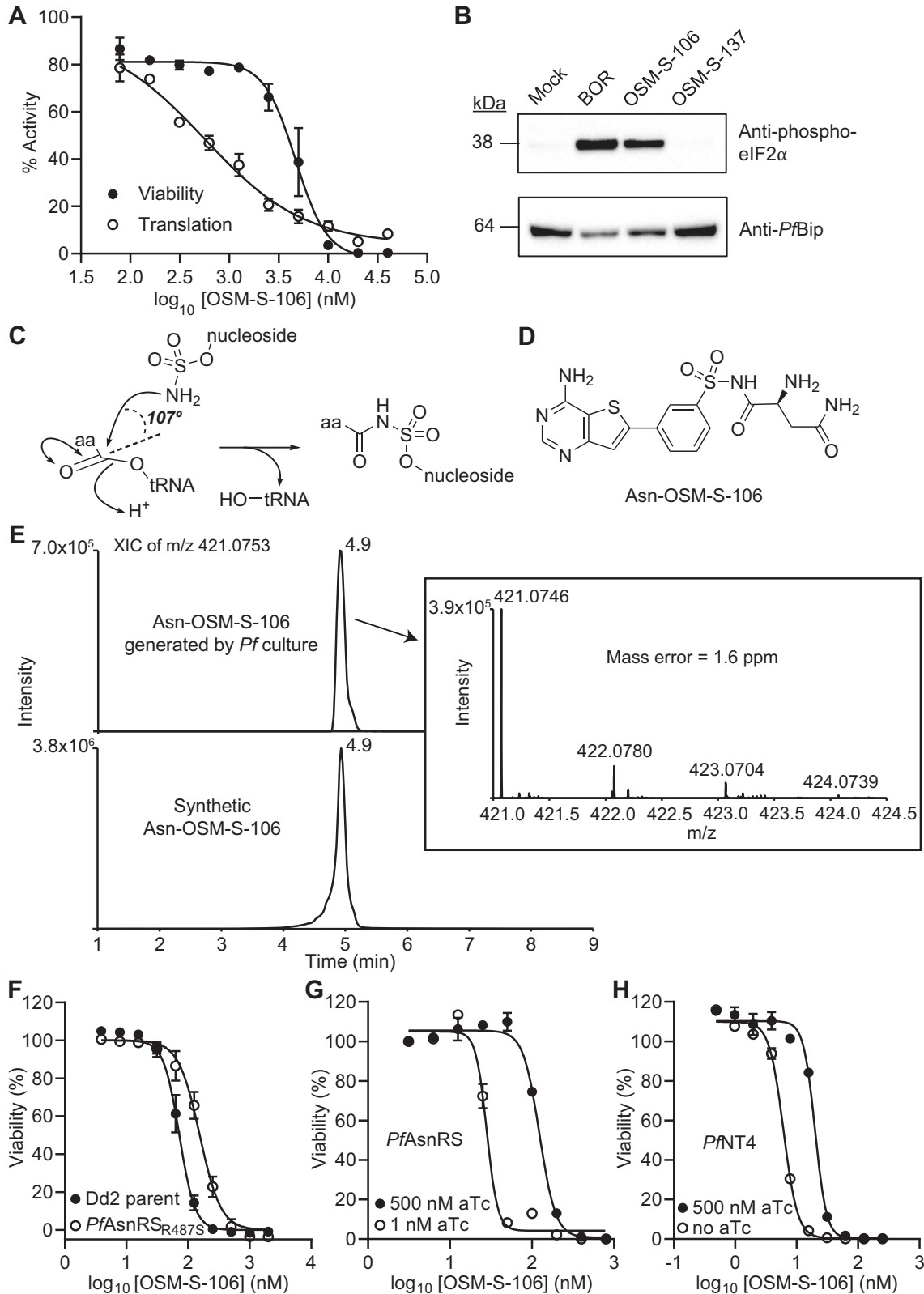

preformed conjugate, as previously observed for ML901 hijacking of *Pf*TyrRS[6].

## Structures of *Hs*AsnRS reveal loop stabilization when Asn-AMP is generated in the enzyme active site

Type II aaRSs typically comprise an N-terminal β-barrel anticodon-binding domain connected via a hinge region to a larger C-terminal catalytic domain that adopts a α–β fold, with three motifs (I–III) involved in ATP binding and dimerization, and a linker domain between motifs II and III[23] (Supplementary Fig. 6). Members of our team previously published a structure of the apo form of an N-terminally truncated *Hs*AsnRS, known as the canonical domain (CD) (PDB: 6AOE[24]). Here, we generated recombinant CD*Hs*AsnRS (A98–P548) and verified that it forms a dimer in solution

**Fig. 2 | Identification of the *P. falciparum* target of OSM-S-106. A** *P. falciparum* cultures (Cam3.II-rev; trophozoite stage; 30–35 h p.i.) were exposed to OSM-S-106 for 6 h. Protein translation was assessed in the last 2 h of the incubation, via the incorporation of OPP. Aliquots of inhibitor-exposed cultures were washed and returned to cultures, and viability was estimated at the trophozoite stage of the next cycle. IC$_{50}$ (Translation) = 0.51 μM, IC$_{50}$ (Viability) = 4.7 μM. Error bars correspond to SEM of three independent experiments. **B** Trophozoite stage Cam3.II_rev parasites (30–35 h p.i.) were incubated with 0.05% DMSO (Mock), 50 nM borrelidin (BOR) or 2.5 μM OSM-S-106 or 2.5 μM OSM-S-137 for 3 h. Western blots of lysates were probed for phosphorylated eIF2α with *Pf*BiP as a loading control. The blot is representative of three independent experiments and additional blots are presented in Supplementary Fig. 1C. **C** Schematic showing aaRS-catalyzed attack of a nucleoside sulfamate on an activated amino acid to form an amino acid adduct.

**D** Structure of Asn-OSM-S-106. **E** *P. falciparum*-infected RBCs were treated with 10 μM OSM-S-106 for 3 h. Extracts were subjected to LCMS. The extracted ion chromatograms of the Asn-OSM-S-106 adduct generated by *P. falciparum* (upper panel) and the synthetic conjugate at *m/z* 421.0753 (lower panel). The inset shows MS analysis of the parasite-generated Asn-OSM-S-106 adduct. **F** Sensitivity to OSM-S-106 exposure (72 h) for a cloned wildtype line (Dd2) and a CRISPR-edited clone harboring *Pf*AsnRS$^{R487S}$. Data represent five replicates and error bars correspond to SD. See Supplementary Table 6 for data values. Sensitivity to OSM-S-106 exposure (72 h) for aptamer-regulatable *Pf*AsnRS (**G**) and *Pf*NT4 (**H**) lines upon addition of aTc (closed circles) and with the target expression reduced (open circles), with data normalized to a no drug control. Data represent the mean of three replicates and error bars correspond to SD. See Supplementary Table 7 for data values.

(Supplementary Fig. 4D, H). We solved the apo structure at a resolution of 1.9 Å, confirming the expected conformation (Supplementary Fig. 7).

Following initial unsuccessful attempts to generate crystals in the presence of ATP and Asn, CD*Hs*AsnRS was incubated in the presence of Asn and the ATP analog, AMPPNP[25]. Diffraction quality crystals were obtained; and we solved the structure (refined at 2.2 Å resolution), revealing the presence of Asn-AMP in the active site, presumably formed by attack of the amino acid on the α-phosphate (Fig. 4A, B and Supplementary Fig. 8). Interactions with the adenylate and asparagine moieties stabilize the activated adenylate in the characteristic bent conformation, with the plane of the ribose angled approx. 90° relative to the adenine ring system, as previously observed in other AsnRSs and indeed other class II synthetases[26,27].

Comparison of the Asn-AMP-bound CD*Hs*AsnRS and our apo CD*Hs*AsnRS structure reveals local changes in and around the active site (Supplementary Figs. 7B and 8B). Of particular interest is residue E279, which lies just N-terminal of the beta hairpin (K286–F295), within a loop that is not well defined in the apo CD*Hs*AsnRS electron density, indicating flexibility. Upon binding of Asn-AMP, the side chain of E279 interacts with the Asn-AMP asparagine moiety, leading to stabilization of the loop. Similarly, upon binding of Asn-AMP, R322 in Motif II interacts with the Asn-AMP phosphate and E324 interacts with the adenylate part of the ligand, leading to stabilization of residues Q325–R329 within a larger loop (Y321–E334) that lies between the two beta strands of Motif II (Fig. 4B and Supplementary Fig. 8B, C). Stabilization of these loops may increase the affinity of binding of the activated intermediate, allowing sufficient residence time for reaction with the cognate tRNA. The conserved "flipping" loop (E279–T283; Supplementary Fig. 6) has previously been shown to undergo dynamic motions that facilitate tRNA binding[28].

### Structures of *Hs*AsnRS in complex with Asn-OSM-S-106 and Asn-AMS

As described above, *Hs*AsnRS is very inefficient in catalyzing the formation of Asn-OSM-S-106. However, we were successful in crystalising CD*Hs*AsnRS in complex with synthetic Asn-OSM-S-106, refined to 2.0 Å resolution (Fig. 4C and Supplementary Fig. 9). OSM-S-106 is located in the adenylate binding pocket; however, in contrast to the ribose of the adenylate-containing structures, the substituted benzene ring of OSM-S-106 is planar with respect to the thienopyrimidine ring system, which positions the amino acid moiety in the correct pose to bind in the same pocket occupied by the asparagine of Asn-AMP (Fig. 4B). Stabilization of the flexible loop structures adjacent to the active site is mediated by side chain interactions of E279 with the asparagine of Asn-OSM-S-106, and interactions of R322 with the sulfonamide group (Supplementary Fig. 9B, C).

We also solved the structure of CD*Hs*AsnRS in complex with synthetic Asn-AMS, refined to 1.9 Å (Supplementary Fig. 10), revealing sodium in the position occupied by magnesium in our Asn-AMP bound

structure. Asn-AMS makes similar interactions with the active site to those observed in the Asn-AMP complex, including the nature of the interactions stabilizing the loops around the active site (Supplementary Figs. 10B, C and 11).

### Sequence alignment and an AlphaFold model of the *Pf*AsnRS structure reveal a Plasmodium-specific insert

Alignment of the *Pf*AsnRS sequence with sequences from a range of species reveals moderate to good conservation (Supplementary Fig. 6). One *Plasmodium*-specific feature of interest is a low complexity insert, adjacent to the flipping loop (Supplementary Fig. 6). In *P. falciparum*, the insert has a length of 76 amino acids[29].

Our attempts to generate a high-resolution crystal structure of *Pf*AsnRS were not successful. We therefore generated a molecular model of the *Pf*AsnRS dimer (Fig. 5A) using AlphaFold Multimer[30]. The model exhibits the anticipated N-terminal β-barrel anticodon-binding domain connected to a larger C-terminal catalytic domain that adopts an α–β fold. The large loop insert is modeled as a partly structured domain that extends from a beta hairpin turn. An overlay of the *Pf*AsnRS model with the CD*Hs*AsnRS structure shows that the long insert interrupts the beta hairpin turn in CD*Hs*AsnRS (Supplementary Fig. 12A). While the insert domain structure is not well-defined, it appears to occupy an area that extends over the active site cavity (Fig. 5A), in a position that could influence the dynamics of the aminoacylation reaction.

### Generation of a *Pf*AsnRS-Asn-tRNA structural model

Our previous studies provided evidence that reaction hijacking involves nucleophilic attack of an aromatic sulfamate/sulfonamide on the amino acid-charged tRNA product in the enzyme active site. For this reaction to occur, the pro-inhibitor needs to bind into the AMP vacated site. We therefore generated a model of the Asn-tRNA-bound *Pf*AsnRS complex (Fig. 5B) to enable docking of different OSM-S-106 derivatives into the AMP-binding site, in the context of the bound Asn-tRNA product. PDB entries for different class II RS enzymes complexed with tRNA were inspected and the *E. coli* AspRS enzyme complexed with Asp-AMP and the cognate *E. coli* tRNA (PDB entry 1C0A)[31], was chosen as a suitable template.

Superimposition of 1C0A onto the AlphaFold model for the *Pf*AsnRS shows that the residues around the active site pocket are closely aligned, except that in the *Pf*AsnRS model the terminal CAA of the tRNA acceptor stem clashes with residues in the flipping loop, a region of the protein known to reposition to allow acceptor stem access, upon tRNA binding[31]. Close alignment of residues lining the active site of *Hs*AsnRS and our model provides further confidence in the active site structure of our *Pf*AsnRS model (Supplementary Fig. 12B, C). The flipping loop residues from 1C0A, which are in the open, acceptor stem-binding conformation were copied into the *Pf*AsnRS model and then manually modified to the correct *Pf*AsnRS sequence. The tRNA from 1C0A was copied into the *Pf*AsnRS model without modification. The Asn-AMP bond was

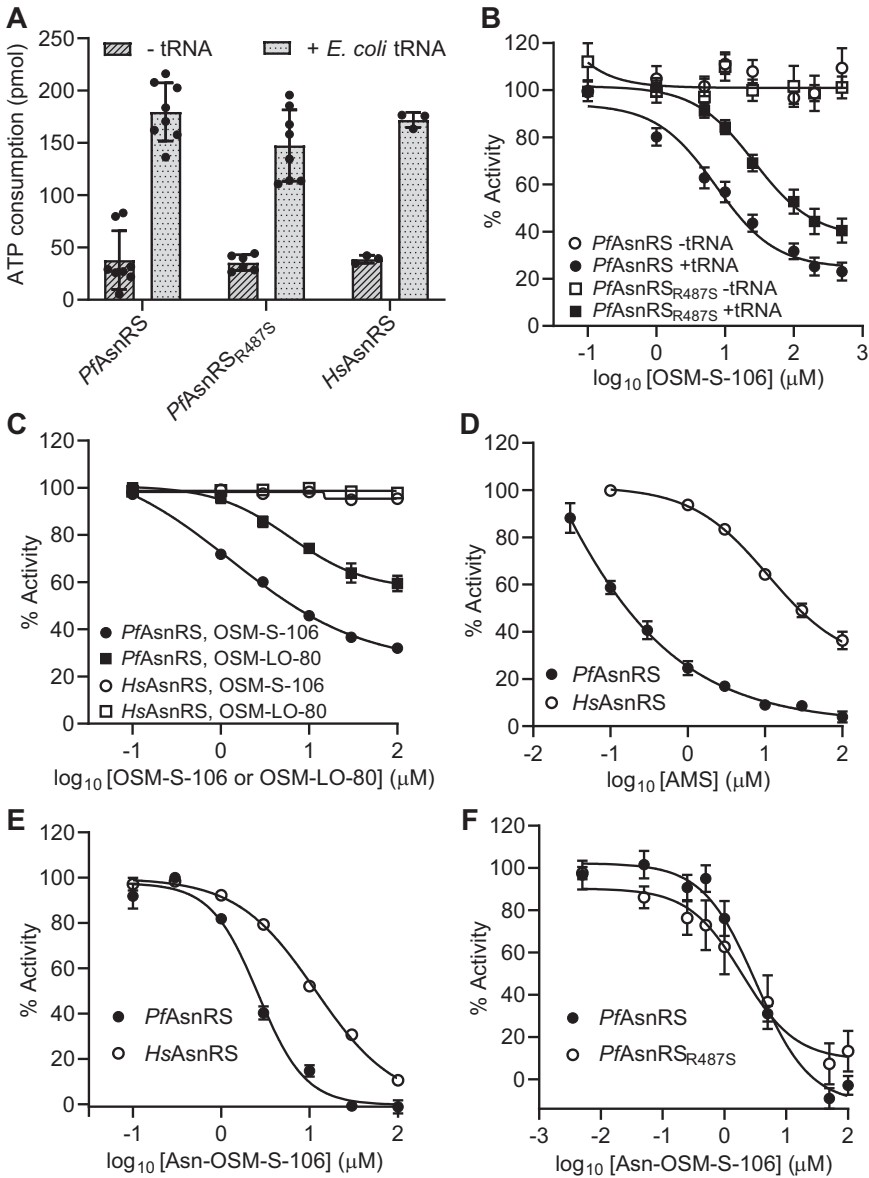

**Fig. 3 | OSM-S-106 hijacks *Pf*AsnRS enzyme activity but is less effective against *Pf*AsnRS_R487 and *Hs*AsnRS. A** ATP consumption by wildtype *Pf*AsnRS, *Pf*AsnRS^R487S and full-length *Hs*AsnRS in the presence and absence of *E. coli* tRNA. Reactions were incubated at 37 °C for 1 h. *Pf*AsnRS and *Pf*AsnRS^R487S: 0.5 μM; *Hs*AsnRS: 0.2 μM. *Pf*AsnRS: *n* = 8; *Pf*AsnRS^R487S: *n* = 6 (−tRNA) and 8 (+*E. coli* tRNA); *Hs*AsnRS: *n* = 3. Error bars correspond to SD. **B** Effects of increasing OSM-S-106 on ATP consumption at 37 °C, over a period of 2.5 h, by wildtype *Pf*AsnRS and *Pf*AsnRS^R487S in the presence or absence of *E. coli* tRNA. Enzyme concentration = 0.5 μM. IC_{50} values: Plus *E. coli* tRNA = 7.3 μM for *Pf*AsnRS and 26 μM for *Pf*AsnRS^R487S; minus *E. coli* tRNA > 500 μM. *Pf*AsnRS: *n* = 6 (−tRNA) and 20 (+*E. coli* tRNA); *Pf*AsnRS^R487S: *n* = 3 (−tRNA) and 15 (+*E. coli* tRNA). Error bars correspond to SEM. **C** Effects of increasing OSM-S-106 and OSM-LO-80 on ATP consumption by *Pf*AsnRS and *Hs*AsnRS. Reactions were incubated at 37 °C for 1 h with 0.05 μM *Pf*AsnRS or 0.2 μM *Hs*AsnRS in the presence of *E. coli* tRNA. IC_{50} values for OSM-S-106: 6.2 μM for

*Pf*AsnRS and >100 μM for *Hs*AsnRS. IC_{50} values for OSM-LO-80: >100 μM for *Pf*AsnRS and *Hs*AsnRS. Data are the average of three independent experiments. Error bars represent SEM. **D** Effects of AMS on ATP consumption by *Pf*AsnRS and *Hs*AsnRS. Reactions were incubated at 37 °C for 1 h with increasing AMS and 0.05 μM *Pf*AsnRS or 0.2 μM *Hs*AsnRS. IC_{50} values: 0.19 μM for *Pf*AsnRS; 26 μM for *Hs*AsnRS. Data represent the average of three independent experiments and error bars correspond to SEM. Effects of synthetic Asn-OSM-S-106 on ATP consumption by *Pf*AsnRS and *Hs*AsnRS (**E**) and *Pf*AsnRS and *Pf*AsnRS_R487S (**F**). Reactions were incubated at 37 °C for 1 or 2.5 h with increasing Asn-OSM-S-106, and 0.5 μM of enzymes without tRNA. IC_{50} values: 2.5/3.3 μM for *Pf*AsnRS; 12 μM for *Hs*AsnRS, 1.9 μM for *Pf*AsnRS_R487S. Data points represent *n* = 3 in (**E**). In (**F**), *n* = 6 (*Pf*AsnRS) and 5 (*Pf*AsnRS^R487S). Error bars correspond to SEM. ATP (10 μM), asparagine (200 μM), pyrophosphatase (1 unit/ml) and *E. coli* tRNA (2.5 mg/ml), if present.

broken and Asn was connected manually to the 3′OH oxygen of the tRNA A76 (Fig. 5B) with AMP remaining in the binding pocket. The modeled complex was then minimized to correct the geometry and remove any steric clashes generated during modeling, using SybylX2.1. Of interest, the residue (R487S) that is modified in *P. falciparum* upon selection for resistance, lies in a helix that partially caps the active site (Fig. 5B) and may interact with the tRNA backbone to stabilize the complex.

## Docking pro-inhibitors into the PfAsnRS-Asn-tRNA model provides a basis for understanding SAR

Susceptibility to reaction hijacking depends on the ability of the enzyme to generate the Asn adduct, rather than the ability to bind the preformed conjugate. Understanding the series SAR therefore requires assessment of suitable, low-energy poses of bound pro-inhibitors for reaction with the Asn-tRNA carbonyl carbon, i.e., a suitable distance and angle[32] between the reacting centers.

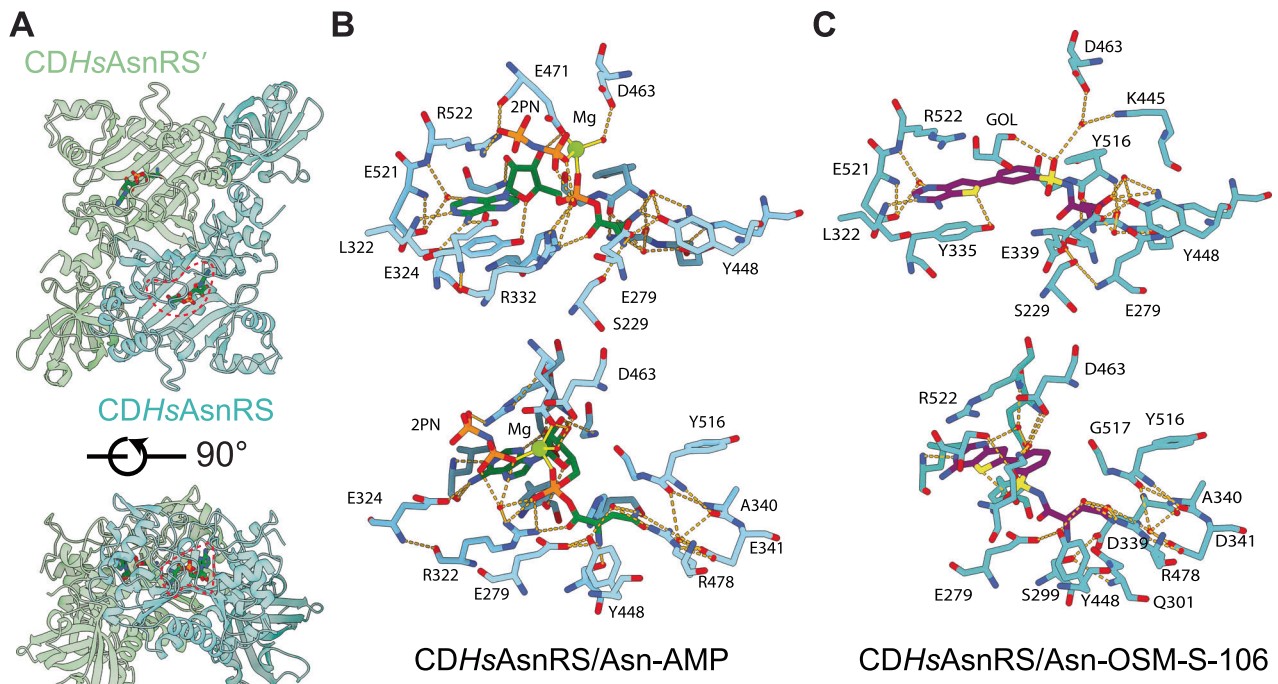

**Fig. 4 | Structures of the CD*Hs*AsnRS/Asn-AMP and CD*Hs*AsnRS/Asn-OSM-S-106 complexes. A** Structure of the CD*Hs*AsnRS dimer in complex with Asn-AMP. The bound Asn-AMP is circled (dotted red lines) and the two chains of the dimer are colored differently. **B** Key inhibitor contact residues in the CD*Hs*AsnRS/Asn-AMP complex. Hydrogen bonds are indicated by yellow dashed lines. **C** Key inhibitor contact residues in the CD*Hs*AsnRS/Asn-OSM-S-106 complex. Hydrogen bonds are indicated by yellow dashed lines. 2PN imidodiphosphoric acid, GOL glycerol. Two orientations of each complex are shown in (**B**) and (**C**).

We first docked AMP and the high potency AMS pro-inhibitor into the AMP-binding site using Surflex in SybylX2.1 (Fig. 5C, D). AMS adopts a very similar position to AMP. Given that there is free rotation around the carbon-sulfur bond, the sulfamate nitrogen can be well positioned to attack the target carbonyl. Importantly, the docking poses of these compounds are similar to the positions of the corresponding components of ligands observed in our structures of the CD*Hs*AsnRS/Asn-AMP and CD*Hs*AsnRS/Asn-AMS complexes. While the AMS pro-inhibitor is a useful positive control compound, its efficient targeting of *Hs*AsnRS makes it a poor starting point for antimalarial drug development. Thus, OSM-S-106 remains of greater interest because of that compound's selectivity.

OSM-S-106 was docked into the AMP pocket of the *Pf*AsnRS-Asn-tRNA complex and adopts a similar conformation in the *Pf*AsnRS model to that observed for the CD*Hs*AsnRS/Asn-OSM-S-106 crystal structure, with the aryl ring twisted toward co-planarity with the thienopyrimidine ring system (Fig. 5E). The model reveals that the sulfonamide nitrogen of OSM-S-106 overlaps with the AMP phosphate and AMS sulfamate, while the thienopyrimidine of OSM-S-106 occupies a similar position to the adenine groups of AMP and AMS. The top scoring docks show rotation of the sulfonamide around the carbon-sulfur bond (Supplementary Fig. 12D), allowing positioning of the OSM-S-106 sulfonamide nitrogen in a good orientation for attack on the Asn-tRNA carbonyl carbon, leading to inhibitor formation.

We compared the docking poses for key OSM-S-106 derivatives. OSM-E-32 has a methyl substitution on the aryl ring, introduced to increase the dihedral angle between the aromatic rings for the purpose of improving solubility. In the lowest energy docked conformation for this compound (Fig. 5F), this ring is twisted relative to the thienopyrimidine ring. Rotation to coplanarity, and adoption of better geometry for the hijacking reaction, would introduce a steric clash between the methyl group and an arginine residue (R584). This inability to adopt a suitable geometry may underpin the decreased antimalarial potency.

OSM-S-488, which has a methyl substituent on the sulfonamide, docks with a pose where the sulfonamide nitrogen is positioned well away from the target carbonyl because the extra methyl group cannot easily be accommodated (Fig. 5G); this poor positioning may underlie its poor activity.

OSM-LO-81 and OSM-LO-80 are sulfamate derivatives of OSM-S-106, adopting the reactive moiety of ML901, with and without an additional carbon in the alkyl linker. In both cases, small changes in distances and geometry arising from steric clashes or inferior low-energy poses appear to equate to large changes in reaction rate (Supplementary Fig. 12E, F).

Similarly, the hydroxyquinazolinyl benzene sulfonamide, OSM-LO-87, exhibited no antimalarial potency. The lowest energy docked conformation for the lowest energy tautomer of OSM-LO-87 shows a substantive shift of the hydroxyquinazolinyl moiety compared with the position for the corresponding thienopyrimidine group of OSM-S-106 (Supplementary Fig. 12G), while for the best docked conformations of the other tautomers, the sulfonamide nitrogen is positioned well away from the target carbonyl. OSM-LO-88 also exhibited very low activity. In many of the lowest energy docks, the oxo-thienopyrimidine adopts a very different position to the amino-thienopyrimidine, with the oxy group pointing in the opposite direction to the amine of OSM-S-106 (Supplementary Fig. 12H). Again, these results suggest that the reactive pose of the pro-inhibitor is important for antimalarial potency, which in these two cases is governed by correct positioning of the thienopyrimidine ring.

Of note is OSM-S-137, a compound with some antimalarial potency and possessing a large substituent on the thienopyrimidine ring. The substituent cannot be accommodated within the AMP pocket and in the lowest energy docked conformation, OSM-S-137 is positioned in the reverse orientation at the active site, relative to OSM-S-106 (Supplementary Fig. 12I). This is consistent with our finding that OSM-S-137 did not induce eIF2α phosphorylation (Fig. 2B);

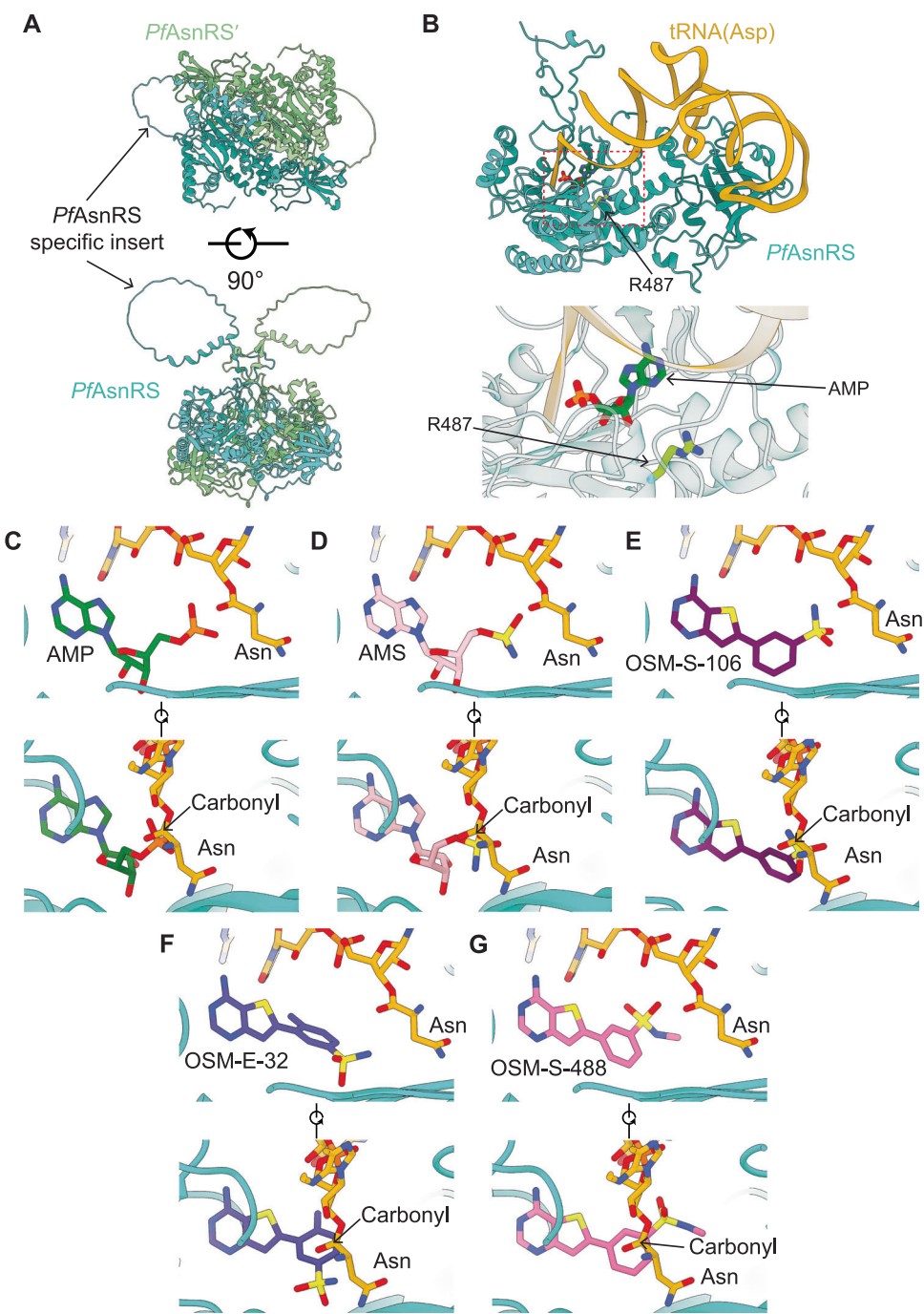

**Fig. 5 | A model of the *Pf*AsnRS-Asn-tRNA complex and compound docking reveal mechanisms for differential compound activity. A** AlphaFold-Multimer model of the *Pf*AsnRS dimer. Each chain of the dimer, and the long, disordered *Pf*AsnRS-specific insert are depicted. **B** Upper panel shows a model of the *Pf*AsnRS-Asn-tRNA complex, generated by overlay of the *Pf*AsnRS model with the *E. coli* AspRS/tRNA(Asp) complex (PDB ID 1C0A[31]). The position of the bound ligand is highlighted with a dotted red line. Residue R487 (arrowed) lies close to the tRNA binding site. Lower panel shows a close-up view of the active site. Representative in silico docks of compounds to the *Pf*AsnRS-Asn-tRNA model for

(**C**) AMP, (**D**) AMS, (**E**) OSM-S-106 (see also Supplementary Fig. 12D), (**F**) OSM-E-32, and (**G**) OSM-S-488. Two orientations of each docked compound are shown to illustrate alignment of the reactive groups with the Asn-tRNA carbonyl carbon. The binding poses of AMP, AMS and OSM-S-106 are similar to the corresponding parts of our experimentally determined structures of the CD*Hs*AsnRS/Asn-AMP, CD*Hs*AsnRS/Asn-AMS, and CD*Hs*AsnRS/Asn-OSM-S-106 complexes. The AMS sulfamate and the OSM-S-106 sulfonamide are in a suitable position to attack the carbonyl carbon of Asn-tRNA.

and suggests that the low-level anti-plasmodial potency of this compound may be off-target.

The modeling described here will be valuable in the design of future OSM-S-106 variants, and potentially in the design of pro-inhibitors of other aaRSs, where suitable distances and geometries are essential in addition to appropriate docking scores.

## Discussion

Nucleoside sulfamates are an exciting class of enzyme pro-inhibitors. They have been shown to induce certain ubiquitin activating (E1) enzymes to synthesize potent inhibitory adducts of the nucleoside sulfamate with enzyme-bound ubiquitin-like proteins. This reaction mechanism is powerful and has resulted in multiple new clinical

candidates that target E1 enzymes (e.g., Pevonedistat, TAK-243 and TAK-981[33–35]). More recently, some aaRSs were found to be susceptible to reaction hijacking by nucleoside sulfamates, in this case, attacking the enzyme bound charged tRNA and forming an inhibitory adduct with the amino acid. The identification of the *Plasmodium*-specific pyrazolopyrimidine sulfamate pro-inhibitor, ML901, opened the possibility of developing bespoke pro-inhibitors that target different *Pf*aaRSs.

ML901 exhibits excellent potency and selectivity, and effects single-dose cure in a humanized mouse model of *P. falciparum* malaria. However, the nucleoside scaffold exhibits low lipophilicity (AlogP of ML901 is 0.069)[6], which may limit its oral bioavailability. Here, we explored an aminothienopyrimidine-based sulfonamide, OSM-S-106, which was first identified in a screen of GSK compounds and has since been explored as part of an Open Source Malaria initiative. OSM-S-106 exhibits drug-like properties with a synthetically accessible scaffold and favorable lipophilicity characteristics (AlogP 1.65). OSM-S-106 exhibits good activity against cultures of *P. falciparum*. Importantly, OSM-S-106 also prevents development of liver stage parasites, suggesting that this class of compound could be used for prophylaxis as well as treatment. One important characteristic of new antimalarial compounds is that they should exhibit a low propensity for resistance. We found that no resistant parasites emerged from an inoculum of $2.4 \times 10^8$ exposed parasites, which compares well with other compounds selected for development[36]. While OSM-S-106 is stable during incubation with human microsomes and rat hepatocytes, it shows rapid intrinsic clearance in mouse microsomes. This complicates studies of pharmacokinetic properties in mice. Here we focused on in vitro analyses.

We used a gradual ramp-up method to evolve resistant parasites with a view to obtaining insights into the target of OSM-S-106. Following 2 months of selection we retrieved parasites exhibiting fourfold resistance that harbored an R487S mutation or a Copy Number Variant (CNV) in cytoplasmic *Pf*AsnRS. We showed that transfectants harboring the *Pf*AsnRS[R487S] mutation have decreased sensitivity to OSM-S-106, while down-regulation of *Pf*AsnRS enhanced sensitivity, validating *Pf*AsnRS as a target.

Interestingly, some of the parasite clones also exhibited mutations in the *P. falciparum* nucleoside transporter 4 (*Pf*NT4). Moreover, down-regulation of *Pf*NT4 enhanced sensitivity to OSM-S-106. Indeed, enhanced sensitivity was observed even in the presence of anhydrotetracycline (aTC), potentially due to low-level down-regulation. A similar base level sensitization has been observed for a *Pf*Hsp70 inhibitor in an aptamer-regulated *Pf*Hsp70 line[37]. *Pf*NT4 is a putative purine transporter that has been shown to be dispensable for blood stage growth but required for sporozoite colonization of salivary glands[20,38,39]. It is possible that *Pf*NT4 transports OSM-S-106 away from its primary site of action and that mutations in *Pf*NT4 enhance the accumulation of OSM-S-106. Further work is needed to test this possibility.

We showed that treatment of cultures with OSM-S-106 inhibits protein translation and triggers eIF2α phosphorylation, which is diagnostic of the presence of uncharged tRNA[18,40], providing further evidence that OSM-S-106 activity leads to a decrease in the level of charged tRNA. If OSM-S-106 indeed inhibits *Pf*AsnRS via a reaction hijacking mechanism, Asn-OSM-S-106 adducts would be generated in the active site. Using targeted mass spectrometry, we detected a strong signal for Asn-OSM-S-106. Of interest, minor MS peaks were also detected for the adducts of glycine and alanine, when cultures were treated with a high concentration of OSM-S-106 (10 μM). This suggests that *Pf*GlyRS and *Pf*AlaRS, both of which are also class II aaRSs, are susceptible to hijacking by OSM-S-106. Our findings that down-regulation of *Pf*GlyRS and *Pf*AlaRS did not enhance susceptibility to OSM-S-106, and that glycine and alanine adducts were not detected in the extracts with 1 μM OSM-S-106 treatment suggest that *Pf*AsnRS is the main target. However, even partial inhibition of *Pf*GlyRS and *Pf*AlaRS may enhance the action of OSM-S-106 and underpin the difficulty of evolving resistance.

We generated recombinant *Pf*AsnRS, *Pf*AsnRS[R487S] and *Hs*AsnRS. ATP consumption by the three enzymes is greatly enhanced by addition of tRNA, consistent with productive aminoacylation. We found that commercially available *E. coli* tRNA was effective as a substrate for all three enzyme preparations, which facilitated the comparison. These biochemical data suggest that the *Pf*AsnRS[R487S] mutation does not affect enzyme activity, consistent with the lack of any obvious growth phenotype.

OSM-S-106 inhibited ATP consumption by *Pf*AsnRS, but only in the presence of tRNA. This is consistent with the reaction hijacking mechanism. The concentration of OSM-S-106 (approx. 1 μM) needed to induce 50% inhibition of ATP consumption is much higher than the amount needed to kill parasite cultures (approx. 60 nM). This may be due to the fact that, in our biochemical assay, the enzyme first needs to generate the charged tRNA product, which is then attacked by the pro-inhibitor to generate the Asn-OSM-S-106 adduct. Tight binding of the adduct prevents the enzyme from undergoing further catalytic cycles. By contrast, in cells, tRNAs are generally fully loaded[41]; and may be able to rebind onto the aaRS, which may promote adduct formation. In addition, sulfonamide- and sulfamate-containing compounds are known to bind to red blood cell carbonic anhydrase[19,42] which may facilitate accumulation of OSM-S-106 into parasitised red blood cells.

Recombinant *Pf*AsnRS[R487S] is less susceptible to inhibition by OSM-S-106 than wildtype *Pf*AsnRS. It is interesting to consider how this mutation might decrease the sensitivity of the enzyme to hijacking by OSM-S-106. Residue R487 lies in a helix that partially caps the active site; and is close to the tRNA binding site. Of interest, analysis of a hybrid structural model of tRNA-bound AsnRS from the filarial nematode *Brugia malayi*[27] revealed an important role for the equivalent residue, R425; this residue is involved in a salt-bridge interaction that needs to be broken to allow access of the 3′ end of tRNA to the active site. Thus, the R487S mutation may affect the stability of the complex of *Pf*AsnRS with the Asn-tRNA product, which may in turn affect residence time of the bound Asn-tRNA and therefore susceptibility to reaction hijacking.

OSM-S-106 does not inhibit ATP consumption by *Hs*AsnRS, and targeted mass spectrometry revealed that *Hs*AsnRS produces very little Asn-OSM-S-106 adduct. A major structural difference between *Pf*AsnRS and *Hs*AsnRS is the presence of a large *Plasmodium*-specific insert, adjacent to the flipping loop. This flipping loop is known to lock the activated Asn-AMP intermediate in place but to 'flip' out of the way to allow the tRNA acceptor stem to insert adjacent to the active site[43]. The presence of the large insert in *Pf*AsnRS may increase the time the Asn-tRNA product remains bound to the enzyme. This may enable AMP to vacate the active site and OSM-S-106 to bind to the site and mount a nucleophilic attack on the susceptible carbonyl group in Asn-tRNA. By contrast, the Asn-tRNA product may be released more rapidly from *Hs*AsnRS, thus limiting the opportunity for reaction hijacking.

Initial attempts to crystallize CD*Hs*AsnRS in the presence of ATP and Asn, with a view to capturing the complex with the activated Asn-AMP intermediate were not successful. Therefore, we employed the ATP analog, AMPPNP, which can serve as a substrate for some aminoacyl tRNA synthetases[25,44]. Diffraction quality crystals were obtained under these conditions. A comparison of the apo and Asn-AMP bound structures reveals stabilization of two flexible regions. A four amino acid stretch between the beta hairpin and Motif I is stabilized by the formation of a contact between E279 and the Asn part of the ligand. Motif II, which lies further toward the C-terminus, is intersected by a second flexible loop. In the presence of bound Asn-AMP, R322 and E324 in Motif II are stabilized by an interaction with the adenylate part of the ligand, further contributing to binding the activated Asn-AMP intermediate.

While *Hs*AsnRS is unable to generate the Asn-OSM-S-106 complex, it is able to bind the synthetic adduct, as evidenced by inhibition of the consumption of ATP in the presence of Asn-OSM-S-106. We generated a high-resolution structure of *Hs*AsnRS in complex with synthetic Asn-OSM-S-106. The OSM-S-106 is positioned in the adenylate binding pocket with the sulfonamide-carbonyl bond overlaying the position of the phosphate-carbonyl bond of Asn-AMP and the sulfamate-carbonyl bond of Asn-AMS. Interestingly, however, the benzene ring of OSM-S-106 lies in the same plane as the thienopyrimidine ring. By contrast, the ribose of the adenylate-containing structures is twisted with respect to the nucleoside.

Our attempts to solve the crystal structure of *Pf*AsnRS at high resolution were not successful, possibly due to the presence of the large insert and an extended N-terminal domain. Previous work has shown that these domains are needed for correct folding of *Pf*AsnRS[29]. We therefore generated a molecular model of *Pf*AsnRS bound to Asn-tRNA, building on structural information for *Hs*AsnRS and *Ec*AspRS/tRNA. The model represents the product-bound form of the enzyme primed for binding of OSM-S-106 or other potential AMP mimics. OSM-S-106 docks into *Pf*AsnRS with the benzene ring twisted toward planarity with the thienopyrimidine ring, in a pose similar to that observed in the crystal structure of *Hs*AsnRS/Asn-OSM-S-106 complex. The position of the sulfonamide nitrogen overlaps with that of the AMP phosphate and the AMS sulfamate nitrogen. Similarly, the amino groups on the pyrimidine rings of OSM-S-106, AMP and AMS all align closely. These docking studies illustrate that potent pro-inhibitors must bind in a manner that precisely positions the reactive sulfonamide to attack the carbonyl carbon of Asn-tRNA.

Interrogation of the lower energy docked conformations of the lower potency OSM-S-106 derivatives reveals a failure to position correctly either the sulfonamide nitrogen or the pyrimidine nitrogen. The work provides insights into the very subtle positioning requirements for reaction hijacking to occur; and provides a basis for the design of new compounds with improved activity.

In summary, this work identifies *Pf*AsnRS as a *P. falciparum* aaRS that can be specifically targeted by reaction hijacking; and identifies OSM-S-106 as an exemplar of a new chemical class of species-specific reaction hijacking inhibitor. The ability to selectively hijack particular aaRSs provides a new way to inhibit a class of enzymes that are considered good drug targets in *Plasmodium* and other infectious organisms[26,45,46]. Our biochemical, structural, and modeling studies reveal the molecular correlates of potent antimalarial activity. This work will help in the development of new, much needed, antimalarial therapies.

## Methods

### Activity against *P. falciparum* cultures

Antimalarial activity against *P. falciparum* 3D7 was tested by TCGLS, Kolkata, India, using the lactate dehydrogenase (*Pf*LDH) growth inhibition assay[7]. Following the 72-h incubation with OSM-S-106 and derivatives, 70 μl of freshly prepared reaction mix containing 143 mM sodium L-lactate, 143 μM 3-acetyl pyridine adenine dinucleotide (APAD), 179 μM Nitro Blue tetrazolium chloride (NBT), diaphorase (2.83 U/ml), 0.7% Tween 20, 100 mM Tris-HCl pH 8.0 was added into each well of the incubation plate. Plates were shaken to ensure mixing and were placed in the dark at 21 °C for 20 min. Data were normalized to percent growth inhibition with respect to positive (0.2% DMSO, 0% inhibition) and negative (mixture of 100 μM chloroquine and 100 μM atovaquone, 100% inhibition) controls. *P. falciparum* strain (3D7) was obtained from BEI Resources.

Alternatively, sorbitol-synchronized parasites (3D7 strain, ring stage)[47] were incubated with OSM-S-106 and other inhibitors for 72 h. Viability was assessed in the second cycle by flow cytometry, following labeling with 2 μM Syto-61 (Thermo Fisher Scientific)[48,49]. Viability represents the parasitemia normalized to untreated and "kill" controls that were treated with 2 μM dihydroartemisinin (DHA; Sigma-Aldrich) for 48–72 h. For drug pulse assays, tightly synchronized Cam3.II-rev[50] parasites (1–1.5% parasitemia, 0.2% final hematocrit) were added to the plates and incubated for 6 h. Drugs were removed and the parasitemia assessed in the trophozoite stage of the next cycle.

### Activity against HepG2 and *P. berghei*

Human hepatic cells ($12 \times 10^3$; HepG2-A16-CD81-EGFP), stably transformed to express a GFP-CD81 fusion), were pretreated for 18 h with decreasing concentrations of the compounds of interest, over the range 50 μM to 0.85 nM. The cells were then infected with freshly dissected luciferase-expressing *P. berghei (PbLuc)* ($4 \times 10^3$) sporozoites[51]. After 48 h of incubation with the compound, the viability of *P. berghei* exoerythrocytic forms (EEF) was measured by bioluminescence using Bright Glow reagent (Promega). HepG2 cytotoxicity was assessed by adding CellTiterGlo reagent (Promega). The plates were read in a PHERAstar FSX reader (BMG LABTECH).

### Metabolic stability study using liver microsomes

A solution of the test compounds in phosphate buffer solution (1 μM) was incubated in pooled human and mouse liver microsomes (0.5 mg/ml) for 0, 5, 20, 30, 45 and 60 min at 37 °C in the presence and absence of an NADPH regeneration system (NRS). The tests were carried out by TCGLS, Kolkata, India. The reaction was terminated with the addition of ice-cold acetonitrile, containing a system suitable standard, at designated time points. The sample was centrifuged ($3300 \times g$) for 20 min at 20 °C and the supernatant was diluted by half in water and then analyzed by LC-MS/MS. The % parent compound remaining, half-life ($T_{1/2}$) and clearance ($CL_{int,app}$) were calculated using standard methodology. The experiment was carried out in duplicate. Verapamil, diltiazem, phenacetin and imipramine were used as reference standards.

### Minimum inoculum of resistance

Minimum inoculum of resistance (MIR) studies were conducted for OSM-S-106 using a modified "Gate Keeper assay"[36]. The $IC_{50}$ was determined to be 88.9 nM ($N, n = 3, 2$), and the $IC_{90}$ was determined to be 169.2 nM ($N, n = 3, 2$) in the *P. falciparum* Dd2-B2 clone. A single-step selection was set up by exposing *P. falciparum* cultures (Dd2-B2, 3% hematocrit; 1E7 Dd2-B2 parasites in each well of a 24-well plate) to $3 \times IC_{90}$ (508 nM) of OSM-S-106 over 60 days. Wells were monitored daily by smear during the first 7 days to ensure parasite clearance, during which media was changed daily. Thereafter, cultures were screened three times weekly by flow cytometry and smearing, and the selection maintained a consistent drug pressure of $3 \times IC_{90}$ over 60 days. No recrudescence was observed over the course of this selection. Control selections with DSM265 (at 58 nM, corresponding to $5 \times IC_{50}$), yielded 14/96 recrudescent wells, consistent with earlier reports[13,36]. Whole-genome sequencing analysis employed MiSeq data from libraries of $2 \times 300$ bp paired end reads[52].

### In vitro evolution of *P. falciparum* with reduced sensitivity to OSM-S-106

*P. falciparum* Dd2 was selected for resistance to OSM-106 over a period of 2 months, starting at the $IC_{50}$ and increasing to $4 \times IC_{50}$. Two independent selections were performed, and two or four clones were isolated from each of the selection flasks by limiting dilution, yielding a total of 6 resistant Dd2 clones. Whole-genome sequencing was applied to an average coverage of 117x. Reads were mapped to the 3D7 reference genome. Mutations that were present in both the resistant clones and their isogenic parent were removed. In addition, the genomes were analyzed for potential copy number variation with the GATK4

CNV pipeline using panels of controls developed for the Dd2 genetic background[53,54].

## Whole-genome sequencing and analysis of OSM-S-106-resistant parasites

The sequencing library for parasite genomic DNA was prepared with the Nextera XT kit (Cat. No. FC-131-1024, Illumina) following the standard dual index protocol. The library was sequenced at the UC San Diego IGM Genomics Center on the Illumina HiSeq 2500 in RapidRun mode to generate 100 bp paired-end reads. Fastq files were aligned to the *P. falciparum* 3D7 reference genome (PlasmoDB v13.0) using the Platypus pipeline[55]. The seven clones generated in the study (one parent clone and six OSM-S-106-resistant clones) were sequenced to an average depth of 132x.

SNVs and INDELs were called against the 3D7 reference genome using GATK HaplotypeCaller and filtered according to GATK recommendations[53]. Briefly, SNVs were retained if they met the following filter criteria: ReadPosRankSum >8.0 or <−8.0, QUAL < 500, Quality by Depth (QD) < 2.0, Mapping Quality Rank Sum <−12.5, and filtered depth (DP) < 7. INDELs were retained if they passed ReadPosRankSum <−20, QUAL < 500, QD < 2, and DP < 7. SnpEff (version 4.3) was used to annotate variants in the resulting VCF file[56]. Variants with passing quality metrics and ≥90% allele frequency were further filtered to remove mutations that were also present in the Dd2 parent clone, as these would not have evolved over the course of OSM-S-106 selection. Each resistant clone contained 3-6 SNVs or INDELs that met all filtering criteria. CNVs were identified by differential Log2 copy ratio[57].

## Generation of a *Pf*AsnRS[R487S] transfectant cell line

A single CRISPR/Cas9 plasmid was used to generate parasites encoding the R487S mutation in *Pf*AsnRS, as shown in Supplementary Fig. 3A−C. Two guide RNAs were designed using Benchling (benchling.com). The gRNA1 (5′-CATTCGAAGTGAAAGTTGAA-3′) and gRNA2 (AGTGAAAGTTGAATGGGGAA) were located near the mutation site. Both gRNAs and their complementary sequences were synthesized as primers by IDT. Each gRNA was cloned into the pDC2-coCas9-gRNA plasmid[58]. A donor template of 780 bp, encompassing coding nucleotide sequences 1045-1824, was synthesized (Thermo Fisher Scientific) and assembled at the *Aat*II and *Eco*RI sites using NEBuilder HiFi DNA Assembly. In addition to the R487S mutation, additional silent shield mutations that prevent Cas9 binding were introduced, as shown in Supplementary Fig. 3C. Transfections were performed on ring-stage Dd2 parasites using a BioRad Gene Pulser II as described[58], with 5 nM WR99210 drug pressure applied for 8 days. Edited clones were isolated by limiting dilution and validated by Sanger sequencing.

## Generation of conditional knockdown parasite lines

Conditional knockdown (cKD) *P. falciparum* lines were generated for the cytosolic AsnRS (PF3D7_0211800), cytosolic AlaRS (PF3D7_1367700), cytosolic GlyRS (GlyRS; PF3D7_1420400), *Pf*NT4 (PF3D7_0103200), *P. falciparum* glutamate dehydrogenase 3 (GDH3; PF3D7_0802000), and *P. falciparum* carbonic anhydrase (CA, PF3D7_1140000) by fusing the coding sequences and non-coding RNA aptamer sequences in the 3′-UTR, permitting translation regulation using the TetR-DOZI system[59,60]. Gene editing was achieved by CRISPR/ *Sp*Cas9 using the linear pSN054 vector that contains cloning sites for the left homology region (LHR) and the right homology region (RHR) as well a target-specific guide RNA under control of the *T7* promoter. Cloning into the pSN054 donor vector was carried out following described procedures[59,60]. The vector includes V5-2xHA epitope tags, a 10x tandem array of TetR aptamers upstream of an *Hsp86* 3′UTR, and a multicistronic cassette for expression of TetR-DOZI (translation regulation), *blasticidin S-deaminase* (selection marker) and a *Renilla luciferase* (*RLuc*) reporter. All primer and synthetic fragment sequences that were generated using the BioXP™ system and IDT gBlocks™ are included in Supplementary Table 9. The final constructs were sequence-verified and further confirmed by restriction digests.

Transfection into Cas9- and T7 RNA polymerase-expressing NF54 parasites was carried out by pre-loading red blood cells (RBCs) with the donor vector[61]. Parasite cultures were maintained continuously in 500 nM anhydrotetracycline (aTc, Sigma-Aldrich 37919) and drug selection with 2.5 µg/ml of Blasticidin S (RPI Corp B12150-0.1) was initiated 4 days after transfection. Cultures were monitored by Giemsa smears and RLuc measurements.

## Growth assay for knockdown parasite lines

Assessment of parasite viability during target protein perturbations was carried out using luminescence as a readout of growth. Synchronous ring-stage parasites, cultured in the presence (50 nM) and absence of aTc, were set up in triplicate in a 96-well U-bottom plates (Corning® 62406-121). Luminescence signals were taken at 0 and 72 h post-invasion using the Renilla-Glo(R) Luciferase Assay System (Promega E2750) and the GloMax® Discover Multimode Microplate Reader (Promega). The luminescence values in the knockdown conditions were normalized to aTc-treated (100% growth) and dihydroartemisinin-treated (500 nM, no growth) samples and results were visualized using GraphPad Prism (version 9; GraphPad Software).

## OSM-S-106 susceptibility assays for knockdown parasite lines

The stock solution of OSM-S-106 was dispensed into 96-well (BD Falcon™ 62406-121) and 384-well (Corning® MPA-3656) U-bottom microplates and serially diluted in complete medium to yield a final concentration in the assay ranging from 0.8−0.003 µM. Synchronous ring-stage *Pf*AsnRS, *Pf*AlaRS, *Pf*GlyRS, *Pf*NT4, *Pf*GDH3, and *Pf*CA cKD parasites, as well as a control line expressing a fluorescent protein under the control of the TetR/DOZI module[59], were maintained in 0.5 µM aTc to achieve wild-type protein levels, and 0.001 or 0.0015 µM aTc for knockdown of *Pf*AsnRS, *Pf*AlaRS and *Pf*GlyRS, and no aTc for knockdown of *Pf*NT4, *Pf*GDH3 and *Pf*CA. DMSO- and dihydroartemisinin-treatment (0.5 µM) served as reference controls. Luminescence was measured after 72 h as described above and IC$_{50}$ values were obtained from corrected dose-response curves using GraphPad Prism.

## Protein translation assay

Highly synchronous *P. falciparum* Cam3.II[rev][50] infected RBCs (30−35 h post-invasion) were exposed to OSM-S-106, cycloheximide, and WR99210 for 4 h. O-propargyl-puromycin (OPP) (Abcam) was added to the culture and incubated for a further 2 h. Parasites were washed three times in 1 × PBS (Gibco™) and fixed with 4% formaldehyde (Polysciences) and 0.02% glutaraldehyde (Sigma) in 1 × PBS for 20 min at room temperature (RT). Cells were washed two times with buffer A (3% human serum in 1 × PBS). Pellets were permeabilized in buffer A containing 0.05% Triton® X-100 and washed two times with buffer A. Fixed-permeabilized cells were subjected to copper-catalyzed azide−alkyne cycloaddition (CuAAC) at 37 °C for 1 h in the presence of 0.1 mM CuSO$_4$, 0.5 mM THPTA, 5 mM sodium ascorbate, and 0.1 µM Alexa Fluor 488 azide in buffer A. Pellets were washed four times in buffer A and resuspended in buffer A containing 25 µg/ml propidium iodide (Invitrogen™). Cells were interrogated by flow cytometry (FACS Canto II; BD Biosciences, San Jose, CA) using FITC and Cy™5.5 channels. Data were collected using BD FACSDiva (version 8.0) and analyzed using FlowJo (version 10.9).

## Western blotting analysis of eIF2α phosphorylation

Highly synchronous *P. falciparum* Cam3.II[rev] infected RBCs (30−35 h post-invasion; 2.5% hematocrit, 5−6% parasitemia) were exposed to OSM-S-106, OSM-S-137, borrelidin (Sigma) or 0.05% DMSO (mock) for 3 h. Infected RBCs were pelleted, washed with ice-cold 1 × PBS +

cOmplete™ EDTA-free protease inhibitor cocktail (Roche), and lysed with 0.03% saponin in 1× PBS on ice. Parasite pellets were washed three times with 1× PBS + cOmplete™ EDTA-free protease inhibitor (Roche) cocktail and centrifuged at 13,000 × g for 10 min. The pellets were solubilized in Bolt™ LDS sample buffer containing reducing agent (Invitrogen™), vortexed at RT for 5 min, and boiled at 95 °C for 5 min. Samples were resolved by SDS-PAGE on Nupage™ 4–12% Bis-Tris acrylamide gel at 150 V for 50 min and transferred to nitrocellulose membranes using iBlot ™ 2 (Life Technologies). Membranes were blocked in PBST (5% (w/v) skim milk in PBS) for 1 h at RT, probed with primary antibodies at 4 °C overnight, and with secondary antibodies at RT for 1 h. Primary antibodies: rabbit anti-phospho-eIF2α (Cell Signaling Technology-119A11; Lot 12 Ref no. 3597L; 1:1000); polyclonal mouse anti-*Pf*BiP (WEHI; 1:1000). Secondary antibodies: goat anti-rabbit IgG-HRP (Chemicon-AP132P; Lot 3584340; 1:20,000); goat anti-mouse IgG-HRP (Chemicon-AP181P; Lot 3557957; 1:50,000). The membranes were washed and incubated with Clarity Western ECL Substrate (Bio-Rad) and imaged using the ChemiDoc ™ MP imaging system (Bio-Rad).

## Mass spectrometry to identify and quantify the OSM-S-106-asparagine conjugate

In vitro AsnRS reactions were set up with the following components: 1 µM *Pf*AsnRS, 20 µM L-asparagine, 10 µM ATP, 10 µM OSM-S-106 and 2.5 mg/ml *E. coli* tRNA (Merck). The reaction buffer consists of 100 mM HEPES pH 7.5 (KOH), 160 mM KCl, 3.5 mM MgCl₂, 1 mM DTT. The mixture was incubated at 37 °C for 1 h. After that, an equal volume of 8 M urea was added to the mixture. Finally, trifluoroacetic acid was added to a final concentration of 1%. The sample was centrifuged at 15,000 × g for 10 min and the supernatant was used for LCMS analysis. Synthetic Asn-OSM-S-106 standards were processed in the same way. Quantification of the conjugates was done using Skyline (version 21.1.0.278).

For identification of conjugates in cell cultures, a late trophozoite stage *P. falciparum* (3D7 strain) culture was exposed to 1 µM or 10 µM OSM-S-106 for 3 h. Following drug treatment, parasite-infected RBCs were lysed with 0.1% saponin in PBS and the parasite pellet was washed 3 times with ice-cold PBS. Cell pellets were kept on ice and resuspended in water as one volume, followed by the addition of five volumes of cold chloroform-methanol (2:1 [vol/vol]) solution. Samples were incubated on ice for 5 min, subjected to vortex mixing for 1 min and centrifuged at 14,000 × g for 10 min at 4 °C to form 2 phases. The top aqueous layer was transferred to a new tube and subjected to LCMS analysis. Data analysis was performed using Xcalibur (version 4.4).

## High-performance liquid chromatography (HPLC) and mass spectrometric (MS) analyses

Samples were analyzed by reversed-phase ultra-high-performance liquid chromatography (UHPLC) coupled to tandem mass spectrometry (MS/MS) employing a Vanquish UHPLC linked to an Orbitrap Fusion Lumos mass spectrometer (Thermo Fisher Scientific, San Jose, CA, USA) operated in positive ion mode. Solvent A was 0.1% formic acid/10 mM ammonium acetate in water and solvent B was 0.1% formic acid/10 mM ammonium acetate in acetonitrile. Ten µl of each sample was injected into an RRHD Eclipse Plus C18 column (2.1 × 1000 mm, 1.8 µm; Agilent Technologies, USA) at 50 °C at a flow rate of 350 µl/min for 3 min using 0% solvent B. During separation, the percentage of solvent B was increased from 0% to 25% in 7 min. Subsequently, the percentage of solvent B was increased to 99% in 0.1 min and then maintained at 99% for 0.9 min. Finally, the percentage of solvent B was decreased to 0% in 0.1 min and maintained for 3.9 min.

MS experiments were performed using a Heated Electrospray Ionization (HESI) source. The spray voltage, flow rate of sheath, auxiliary and sweep gases were 3.5 kV, 20, 6, and 1 'arbitrary' unit(s),

respectively. The ion transfer tube and vaporizer temperatures were maintained at 350 °C and 400 °C, respectively, and the S-Lens RF level was set at 50%. A full-scan MS spectrum and targeted MS/MS for proton adduct of Asn-OSM-S-106 or 20 possible common amino acid-containing inhibitor adducts were acquired in cycles throughout the run. The full-scan MS-spectra were acquired in the Orbitrap at a mass resolving power of 120,000 (at *m/z* 200) across an *m/z* range of 200–1500 using quadrupole isolation and the targeted MS/MS were acquired using higher-energy collisional dissociation (HCD)-MS/MS in the Orbitrap at a mass resolving power of 7500 (at *m/z* 200), a normalized collision energy (NCE) of 20% and an *m/z* isolation window of 1.6.

## Analytical ultracentrifugation

*Pf*AsnRS, *Pf*AsnRS$_{R478S}$, CD*Hs*AsnRS and *Hs*AsnRS samples were diluted to 2.8 µM in 25 mM Tris-HCl, pH 7.4, 150 mM NaCl and 0.5 mM TCEP. Four hundred µl aliquots were loaded into double-channel quartz window cells (Beckman Coulter), with the above buffer in the reference compartment. Cells were centrifuged at 201,600 × g or 129,024 × g at 20 °C using an XL-I analytical ultracentrifuge (Beckman Coulter) or an Optima analytical ultracentrifuge (Beckman Coulter). Radial absorbance data were acquired at a wavelength of 238 or 280 nm (as indicated), with radial increments of 0.003 cm, in continuous scanning mode. The sedimenting boundaries were fitted to a model that describes the sedimentation of a distribution of sedimentation coefficients with no assumption of heterogeneity (c(s)) using the program SEDFIT (version 16.1c)[62]. Data were fitted using a regularization parameter of *p* = 0.95, floating frictional ratios, and 250 sedimentation coefficient increments.

## ATP consumption assay

The consumption of ATP by wildtype *Pf*AsnRS, *Pf*AsnRS$^{R487S}$ and *Hs*AsnRS was determined using a luciferase-based assay as per the manufacturer's instructions (Kinase-Glo® Luminescent Kinase Assay, Promega). Reactions were conducted in 100 mM HEPES pH 7.5, 160 mM KCl, 3.5 mM MgCl₂, 0.1 mg/ml BSA, 1 mM DTT, with 200 µM L-asparagine, 10 µM ATP, 1 unit/ml inorganic pyrophosphatase and 2.5 mg/ml *E.coli* tRNA (if present). Enzyme concentration and incubation time for each experiment are described in the figure legends. Reactions were incubated at 37 °C, followed by addition of the Kinase-Glo reagent and incubation for 10 min at room temperature. Luminescence output was measured using a plate reader (CLARIOstar, BMG LABTECH) and MARS data analysis software (version 3.32). The concentration of ATP was quantified by linear regression using an ATP standard curve (Microsoft Excel). Data are normalized to the ATP consumption by DMSO (0.5%) treated AsnRS as a positive control (100% activity). Samples with no enzyme served as negative controls. Dose-response curves and IC$_{50}$ values were obtained using GraphPad Prism.

## Expression and purification of His-tagged human AsnRS canonical domain (His-CD*Hs*AsnRS)

The expression and purification of His-tagged human AsnRS canonical domain, residues A98–P548 (His-CD*Hs*AsnRS) has been described[24]. Briefly, the amino acid sequence comprising residues A98-P548 with N-terminal His₆-tag was expressed via pET-28a in *E. coli* strain Solu_BL21 (Genlantis). Cells were cultivated in 1 L LB media supplemented with 50 µg/ml ampicillin in a shaker-incubator at 37 °C to OD$_{600}$ = 0.5. Recombinant protein expression was induced by addition of 0.5 mM isopropyl β-D-1-thiogalactopyranoside (IPTG). Cultures were further incubated for 4 h at 37 °C and cells harvested by centrifugation (6000 × g). Pelleted cells were resuspended in lysis buffer containing 0.5 M NaCl, 20 mM Tris-HCl (pH 7.5), 35 mM imidazole, and 1 mM β-mercaptoethanol, lysed with an ultrasonic processor (Cole-Parmer), and centrifuged at 35,000 × g for 30 min. The supernatant

was filtered with 0.45-μm syringe filter device (Sartorius) and loaded onto a HisTrap chelating 5-ml HP column (Cytiva). The loaded column was washed with lysis buffer, and retained His-CDHsAsnRS was eluted with an increasing gradient of lysis buffer containing 1 M imidazole. Prior to ion-exchange chromatography, fractions containing CDHsAsnRS were buffer-exchanged with binding buffer; 100 mM NaCl, 20 mM Tris-HCl pH 7.5, and 5 mM dithiothreitol using a HiPrep desalting 26/10 column (Cytiva) and loaded onto a HiTrap Q 5-ml HP column (Cytiva). CDHsAsnRS was eluted with an increasing gradient of binding buffer containing 1 M NaCl and finally subjected to a HiLoad 16/600 Superdex 200 pg column (Cytiva) equilibrated with the buffer containing 200 mM NaCl, 10 mM HEPES-NaOH (pH 7.0).

### Expression and purification of native PfAsnRS, PfAsnRS$_{R478S}$, HsAsnRS and CDHsAsnRS recombinant proteins

Plasmid vectors were designed to express recombinant PfAsnRS, PfAsnRS$_{R478S}$, HsAsnRS (residues M1–P548) and CDHsAsnRS (residues A98–P548) comprising a hexa-histidine tag at the N-terminus, an intervening TEV cleavage sequence and C-terminal AsnRS sequence (His-TEV-AsnRS). Open reading frames were codon optimized for expression in E. coli, synthesized and cloned into the pET11a expression vector (GeneScript). E. coli BL21 (DE3) containing the expression vector were cultivated in 2 L LB media containing 100 μg/ml ampicillin in a shaker-incubator at 37 °C. The culture was transferred to a 16 °C shaker-incubator when the cell density approached mid log phase (OD$_{600}$ approx. 0.6). Recombinant His-TEV-AsnRS expression was induced by addition of 0.1 mM IPTG to the culture medium and the cells were incubated for an additional 16 h. Cells were harvested by centrifugation (6000 × g) and resuspended in 40 ml lysis buffer containing 50 mM Tris-HCl, pH 7.4, 350 mM NaCl, 40 mM imidazole, 0.5 mM TCEP, 1 mg/ml lysozyme and 1x protease inhibitor cocktail (Roche). Cells were lysed by sonication (Microtip, QSonica) and the lysate clarified by centrifugation at 30,000 × g for 25 min at 4 °C and passage through 0.8/0.2 μm (Pall) syringe filter. The supernatant was applied to a 5 ml HisTrap HP column (GE Healthcare) and washed with 50 ml binding buffer containing 50 mM Tris-HCl, pH 7.4, 350 mM NaCl, 40 mM imidazole, and 0.5 mM TCEP. His-TEV-AsnRS enzyme was eluted using a 0–500 mM imidazole gradient in binding buffer over 100 ml. His-tagged TEV protease (L56V/S135G/S219V triple-mutant[63]) was added to His-TEV-AsnRS (mass ratio 1:100, His-TEV-AsnRS:His-TEV protease) and dialyzed overnight at 4 °C against 50 mM Tris-HCl, pH 7.4, 350 mM NaCl, 40 mM imidazole, 0.5 mM TCEP. The resultant native AsnRS enzyme was isolated from cleaved His-tag and His-TEV protease by passage of sample through a 5 ml HisTrap HP column and collection of flow-through material. Native AsnRS enzyme was further purified by gel filtration using a HiLoad 16/600 Superdex 200 column (GE Healthcare), pre-equilibrated in 25 mM Tris, pH 7.4, 150 mM NaCl and 0.5 mM TCEP.

### Crystallization and X-ray diffraction data collection

For crystallization of Asn-AMP-bound CDHsAsnRS, purified apo His-CDHsAsnRS was concentrated to 10 mg/ml. Crystals were first obtained with a solution containing 20% (v/v) glycerol, 40 mM potassium phosphate, and 16% (w/v) polyethylene glycol 8000 using the hanging drop vapor diffusion method at 295 K. The drops containing crystals were mixed with the same volume of their reservoir solutions supplemented with 10 mM adenylyl imidodiphosphate (AMP-PNP) lithium salt hydrate, 10 mM L-asparagine, and 20 mM MgCl$_2$. The crystals were further incubated for 8 h at 295 K, flash-cooled in a 100 K nitrogen stream, and subjected to X-ray diffraction at the Korean Synchrotron: PAL/PLS BEAMLINE 5C. The collected data were processed with HKL2000[64]. Initial phase estimates were obtained by molecular replacement with PHASER using the previous apo-His-CDHsAsnRS structure (PDB ID: 5XIX) as a template. Automated structure refinement using phenix.refine[65] was followed iteratively by

manual model building in COOT[66]. The statistics for the Asn-AMP-bound CDHsAsnRS structure are shown in Supplementary Table 8.

For crystallization of Asn-OSM-S-106-bound, Asn-AMS-bound and apo CDHsAsnRS, the protein samples were concentrated to 10 mg/ml. Crystals were obtained in a solution containing 20% (v/v) glycerol, 40 mM potassium phosphate and 14% polyethylene glycol 8000, and 100-mM Tris pH 7.6 using the sitting drop vapor diffusion method at 295 K. Drops containing crystals were mixed with their reservoir solutions supplemented with 350 μM Asn-OSM-S-106 or 350 μM Asn-AMS. The crystals were further incubated for 24 h at 295 K. Crystals were flash-cooled in liquid nitrogen directly from the crystallization drop, and X-ray diffraction data were collected at 100 K and a wavelength of 0.9537 Å using the Eiger 16M detector at the MX2 beamline of the Australian Synchrotron[67]. Diffraction data were indexed and integrated using XDS[68] and analyzed using POINTLESS[69], prior to merging by AIMLESS[70] from the CCP4 software suite (version 8.0)[71]. Initial phase estimates were obtained by molecular replacement in PHASER[72] using modified coordinates of our Asn-AMP-bound CDHsAsnRS as the search model. Automated structure refinement using phenix.refine[65] was followed iteratively by manual model building in COOT[66]. Structure refinement was performed using translation/libration screw (TLS) refinement with each chain comprising a single TLS group. Restraints for Asn-OSM-S-106 and Asn-AMS were generated using phenix.elbow[73]. The identity of the bound metal ion, coordination sphere, and distances were validated for Asn-AMS bound CDHsAsnRS using Check My Metal[74]. Composite omit maps were generated using Phenix (version 1.19.2). Final data collection and refinement statistics are shown in Supplementary Table 8.

### Modeling of the P. falciparum AsnRS-Asn-tRNA complex

A model of the PfAsnRS-Asn-tRNA complex was generated by combining a modified version of the AlphaFold model for PfAsnRS bound to Asn-AMP with the tRNA from the structure of the E. coli aspartyl-tRNA synthase/tRNA complex, 1C0A[31]. The catalytic domain of the PfAsnRS model was aligned to the equivalent region of 1C0A using PyMOL[75] and visual inspection showed an extremely good match for the local structure, with the tRNA from 1C0A positioned appropriately across both the active site and onto the anticodon domain. The only significant clash was of the acceptor stem with residues of the flipping loop adjacent to the active site, due to the PfAsnRS model having these in the closed conformation seen in the tRNA-free structures of class II tRNA synthase enzymes[31]. The conformation of the flipping loop in the PfAsnRS model was manually corrected to the open position using COOT (version 0.9.8.1)[66], and the PfAsnRS/Asn-AMP/tRNA complex model was minimized to remove any minor steric overlaps using SybylX2.1 (Certara, NJ, USA). To generate the PfAsnRS/AMP/Asn-tRNA complex, the bond between the asparagine residue and AMP was manually broken and a new bond to the 3'OH oxygen of the acceptor stem terminal adenine was added using SybylX2.1. The modified complex was minimized to correct any errors in bond lengths or angles.

### Chemistry materials and methods

Chemistry synthetic protocols are provided in Supplementary Information.

### Reporting summary

Further information on research design is available in the Nature Portfolio Reporting Summary linked to this article.

## Data availability

Additional data are available in Supplementary Information. The following structures have been deposited in the PDB: HsAsnRS/Asn-AMP −PDB 8H53; HsAsnRS (apo)−PDB 8TC7; HsAsnRS/Asn-AMS−PDB 8TC8; HsAsnRS/Asn-OSM-S-106−PDB 8TC9. Source data are provided with this paper.

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

## Acknowledgements

We thank Sally-Ann Poulsen, Griffith University and Lawrence R. Dick, Seofon Consulting, for useful discussions. We thank a number of contributors to early research towards understanding the mechanism of action as part of Open Source Malaria, including Rafael M. Couñago and Caio Vinicius dos Reis (formerly UNICAMP, now University of North Carolina, Chapel Hill) for running an early version of an AsnRS biochemical assay, Stuart Ralph (University of Melbourne), Joe DeRisi and Valentina Garcia (UCSF) for advice related to a cell-free protein translation assay, Shozeb Haider (UCL) and Ho Leung Ng (Kansas State University) for an early version of an AsnRS homology model and Anthony Sama (citizen scientist), Chris Swain (Cambridge MedChem Consulting). We thank Paul Willis, Benoit Laleu and Delphine Baud, Medicines for Malaria Venture, Switzerland, and Sabine Ottilie, UC San Diego, for technical support and advice. We thank staff at TCG LifeSciences, Kolkata, India, the antimalarial screening team for 3D7 parasite assays and Jayatri Naskar, Priyam Sen, Sayan Kr Saha for microsomal assays. We thank Sue Charman (MIPS, Monash University) for help with initial microsomal assays. AMS was kindly provided by Steven Langston, Drug Discovery Sciences, Takeda Pharmaceuticals International Company, Cambridge, United States. We would like to thank the Melbourne Mass Spectrometry and Proteomics Facility and Yee-Foong Mok, Melbourne Protein Facility, The Bio21 Molecular Science and Biotechnology Institute, University of Melbourne, for technical support. This research was partly undertaken at the Australian Synchrotron, part of the Australian Nuclear Science and Technology Organization, and made use of the ACRF Detector on the MX2 beamline, and partly at BL-5C of the Pohang Accelerator Laboratory (Pohang, Republic of Korea). We thank the beamline staff for their assistance. We acknowledge facilities provided through the Malaria Drug Accelerator, supported by the Bill and Melinda

Gates Foundation. We would like to acknowledge funding from the Australian National Health and Medical Research Council (APP2022075; to L.T.), the Australian Research Council (LP120100552; to M.H.T.), Medicines for Malaria Venture (MMV; to L.T., M.H.T. and D.A.F.), the Wellcome Trust (206194/Z/17/Z; to M.C.S.L.), the Wellcome Sanger Institute (to M.C.S.L.). We acknowledge the National Research Foundation of Korea (NRF) grant funded by the Korea government (MSIT) (grant number: 2019R1A2C1090251 and RS-2023-00218543 to B.W.H.) and the Korea Health Technology R&D Project through the Korea Health Industry Development Institute (KHIDI), funded by the Ministry of Health & Welfare, Republic of Korea (grant number: HP23C0102 to B.W.H.). E.A.W., D.A.F., C.F.A.P., M.R.L., E.S.I., J.S.P., M.L.S., K.J.F., T.Y., D.E.G., M.C.S.L., K.K., L.C.G. and S.D. are members of the Malaria Drug Accelerator and are supported by a grant to EAW from the Bill and Melinda Gates Foundation (OPP1054480). M.R.L. was supported in part by a Ruth L. Kirschstein Institutional National Research Award from the National Institute for General Medical Sciences, T32 GM008666. L.T. was supported by an Australian Research Council Laureate Fellowship.

## Author contributions

Conceptualization: S.C.X., Y.W., C.J.M., D.E.G., J.B., D.A.F., M.C.S.L., E.A.W., M.D.W.G., M.H.T. and L.T.; Investigation: S.C.X., Y.W., C.J.M., R.D.M., C.D., C.F.A.P., E.D., T.P., M.R.L., M.L.D.S., J.L.S.-N., K.K., E.S.I., D.M.K., M.N.B., J.S.P., K.J.F., T.Y., L.C.G., S.D., O.Y., N.K., Y.D., M.A., G.S., S.N., N.W., D.S.T., B.W.H., G.P.J., B.C.M. and D.L.; Analysis: S.C.X., Y.W., C.J.M., R.D.M., C.D., C.F.A.P., E.D., M.R.L., M.L.D.S., J.L.S.-N., K.K., E.S.I., D.M.K., M.N.B., J.S.P., K.J.F., T.Y., L.C.G., S.D., O.Y., P.J.R., N.K., S.N., N.W., D.S.T., B.W.H., G.P.J., B.C.M., P.T., T.F., J.C.N., D.E.G., J.B., D.A.F., M.C.S.L., E.A.W., M.D.W.G., M.H.T. and L.T.; Funding acquisition: S.C.X., D.E.G., J.B., D.A.F., M.C.S.L., E.A.W., M.D.W.G., M.H.T. and L.T.; Writing: S.C.X., Y.W., C.J.M., C.F.A.P., D.E.G., J.B., D.A.F., M.C.S.L., E.A.W., M.D.W.G., M.H.T. and L.T.

## Competing interests

The authors declare no competing interests.

## Additional information

Stanley C. Xie [1,22], Yinuo Wang [2,22], Craig J. Morton [3,22], Riley D. Metcalfe [4,22], Con Dogovski[1], Charisse Flerida A. Pasaje [5], Elyse Dunn [1], Madeline R. Luth[6], Krittikorn Kumpornsin[7,8], Eva S. Istvan [9], Joon Sung Park [10], Kate J. Fairhurst[11,12], Nutpakal Ketprasit [1], Tomas Yeo[11,12], Okan Yildirim[13], Mathamsanqa N. Bhebhe[14], Dana M. Klug[2], Peter J. Rutledge [14], Luiz C. Godoy [5], Sumanta Dey[5], Mariana Laureano De Souza [6], Jair L. Siqueira-Neto[6], Yawei Du [1], Tanya Puhalovich [1], Mona Amini[1], Gerry Shami [1], Duangkamon Loesbanluechai[7], Shuai Nie[15], Nicholas Williamson [15], Gouranga P. Jana[16], Bikash C. Maity[16], Patrick Thomson [17], Thomas Foley[17], Derek S. Tan[13], Jacquin C. Niles[5], Byung Woo Han [10], Daniel E. Goldberg [9], Jeremy Burrows[18], David A. Fidock [11,12,19], Marcus C. S. Lee[7,20], Elizabeth A. Winzeler[6,23] ✉, Michael D. W. Griffin [1,23] ✉, Matthew H. Todd [2,21,23] ✉ & Leann Tilley [1,23] ✉

[1]Department of Biochemistry and Pharmacology, Bio21 Molecular Science and Biotechnology Institute, The University of Melbourne, Melbourne, VIC 3010, Australia. [2]School of Pharmacy, University College London, London WC1N 1AX, UK. [3]Biomedical Manufacturing Program, CSIRO, Clayton South, VIC, Australia. [4]Center for Structural Biology, Center for Cancer Research, National Cancer Institute, Frederick, MD 21702, USA. [5]Department of Biological Engineering, Massachusetts Institute of Technology, Cambridge, MA 02139, USA. [6]Department of Pediatrics, School of Medicine, University of California, San Diego, La Jolla, CA 92093, USA. [7]Parasites and Microbes Programme, Wellcome Sanger Institute, Hinxton CB10 1SA, UK. [8]Calibr, Division of the Scripps Research Institute, La Jolla, CA 92037, USA. [9]Division of Infectious Diseases, Department of Medicine, Washington University in St. Louis, St. Louis, MO, USA. [10]Research Institute of Pharmaceutical Sciences and Natural Products Research Institute, College of Pharmacy, Seoul National University, Seoul 08826, Republic of Korea. [11]Center for Malaria Therapeutics and Antimicrobial Resistance, Columbia University Medical Center, New York, NY 10032, USA. [12]Department of Microbiology and Immunology, Columbia University Medical Center, New York, NY 10032, USA. [13]Chemical Biology Program, Sloan Kettering Institute, Memorial Sloan Kettering Cancer Center, New York, NY 10065, USA. [14]School of Chemistry, University of Sydney, Camperdown, NSW 2006, Australia. [15]Melbourne Mass Spectrometry and Proteomics Facility, Bio21 Molecular Science and Biotechnology Institute, The University of Melbourne, Melbourne, VIC 3010, Australia. [16]TCG Lifesciences Private Limited, Salt-Lake Electronics Complex, Kolkata, India. [17]School of Chemistry, The University of Edinburgh, Edinburgh EH9 3JJ, UK. [18]Medicines for Malaria Venture, 20, Route de Pré-Bois, 1215 Geneva 15, Switzerland. [19]Division of Infectious Diseases, Department of Medicine, Columbia University Medical Center, New York,

NY 10032, USA. [20]Wellcome Centre for Anti-Infectives Research, Biological Chemistry and Drug Discovery, University of Dundee, Dundee DD1 4HN, UK. [21]Structural Genomics Consortium, University College London, London WC1N 1AX, UK. [22]These authors contributed equally: Stanley C. Xie, Yinuo Wang, Craig J. Morton, Riley D. Metcalfe. [23]These authors jointly supervised this work: Elizabeth A. Winzeler, Michael D. W. Griffin, Matthew H. Todd, Leann Tilley. ✉e-mail: ewinzeler@health.ucsd.edu; mgriffin@unimelb.edu.au; matthew.todd@ucl.ac.uk; ltilley@unimelb.edu.au

