## [Peer Review File · Nature Communications]

Reaction hijacking inhibition of Plasmodium falciparum asparagine tRNA synthetaseREVIEWER COMMENTS

Reviewer #1 (Remarks to the Author):

The authors have discovered novel inhibitors of Aminoacyl-tRNA synthetases that form covalent adducts with the aminoacids to treat malaria. Previously discovered compounds were based on sulfamate compounds -OSO₂-NH₂. This led to the view that -OSO₂NH₂ is nucleophilic enough to react with the AA-C(=O)-O-tRNA. However it was not clear if sulfonamides R-SO₂NH₂ can undergo similar reactions. IN this paper the authers have expanded their previous approach and disclose sulfonamides as inhibitors of Aminoacyl-tRNA synthetases that form covalent adducts with AA. Because many drugs contain sulfonamides this paper has important implications for the entire field of drug discovery since it will be very important to test any sulfonamide drugs for its ability to form covalent adducts with various electrophiles in the cells, since the resulting conjugates can contribute to the observed pharmacological activity of sulfonamides. The paper is well written and can be accepted as is.

Reviewer #2 (Remarks to the Author):

The manuscript by Xie et al. makes important contributions to the design of new therapeutics for Plasmodium. A few important scholarly and technical concerns must be addressed before accepting the manuscript.

As the reviewer was not familiar with the field of synthetic chemistry, no comments were made for the "Synthetic Procedure" section. This section may be reviewed by an appropriate researcher/reviewer.

1. Can aminoacyl-tRNA synthetases hydrolyse AMPPNP?
Are there any studies reported on AMPPNP hydrolysis by aaRSs or other enzymes? The supporting references should be included.
2. ATP may have been used and therefore an attempt should be made to model pyrophosphate (DPO) instead of imidodiphosphoric acid (2PN) and B-factor/occupancy can be compared.
3. The environment of the bound ion in the HsAsnRS/Asn-AMS structure indicating that this ion may be Mg²⁺ instead of Na ion. The coordination sphere/distance should be used to validate the bound ion.
4. The difference Fourier/Omit maps for the bound ligands should be included to verify unequivocally bound ligands. The above maps may be added for all four molecules (A-D) in the HsAsn/ASN-AMP structure.
5. Do all four molecules (A-D) in the ASU have the same conformation? particularly for active site residues, and Motif II in the different ligands bound structures.
6. A structural overlay of molecules A-D for different ligand bound forms may be included. The active site residues & Motif II can be highlighted. This will allow to see if there are any noticeable or significant changes between the molecules.
7. The ligand is not clearly visible in the figures of cartoon representation (Fig. 4A, Figs. S8, S9 and S10) and therefore the ligand may be drawn as space-filling model or molecular surface. Several values should be corrected in the crystallographic table.

8. There is no consistency in protein/ligand nomenclature. This should be maintained consistently throughout the manuscript (text, Figure legends and Tables). AMP-Asn has been used in some places instead of Asn-AMP. These needs to be fixed

Minor comments:

Asn-AMP/HsAsnRS (Page 12 and Figure S8 legend)
Asn-AMS/HsAsnRS (Figure S10 legend)
HsAsnRS/Asn-AMP
HsAsnRS/Asn-AMS
HsAsnRS/Asn-OSM-S-106 (Page 16 and Page 17)
Asn-OSM-S-106/HsAsnRS ((Figure S9 legend)
HsNRS/Asn-AMP (Figure 4)
AMP-Asn (Page 11 and Page 19)

Both Figure or Fig. notation have been used and these should be corrected.

"-" hyphen should be added to asparaginyl tRNA synthetase and aminoacyl tRNA synthetases

P. falciparum tyrosine RS (PfTyrRS) can be wrtten as *P. falciparum* tyrosine-tRNA synthetase (PFTyrRS).

delete "-" hyphen in *Homo sapiens* Tyr-RS (HsTyrRS)

The label notation of the mutant residue can be added to superscript PfAsnRSR487S instead of subscript PfAsnRSR487S

Page 4:

Some IC50 values are with SD and some values are without SD

4.4 ± 2.6 μM

2.1 μM, 0.9 μM

Check the statement -- Conversion of the sulphonamide to a sulfamate (OSM-LO-80; Fig. 1E; 2.1 μM) or a sulfamate with an extended linker (OSM-LO-81; Fig. 1F; 0.9 μM) decreased activity (Table 1).

The activity is increased for OSM-LO-81 but statement is reporting that activity decreased. This needs to be fixed.

Page 5:

Abbreviations for SNV and CNV can be added where they first appear

IC90 values should be corrected

3 x IC90 (507.7 nM) Page 5

3 x IC90 (508 nM) Page 18

The abbreviation of dhodh should be mentioned.

More clarity is required in the following sentence.

"This yielded 14 recrudescence wells (corresponding to a Minimum Inoculum for Resistance (MIR) of 1.4×10^6), with evidence of separate amplification events encompassing the dhodh locus consistent with increased IC50 values (Supplementary Table 3,4; Supplementary Dataset 2)."

"This selection yielded 21 newly emerged coding variants in 10 unique core genes (Table 2; Supplementary Table 5; Supplementary Datasets 3,4).

Page 6:

"-" hyphen should be deleted with a blank space - 6-h exposure

The prolyl-tRNA synthetase inhibitor halofuginone also induces eIF2 α phosphorylation. This can be mentioned with reference

Page 8:

Mujumdar et al., 2018 is not in the reference list. The number system should be used for this reference invoking

Page 9 and Page 10:

Can aminoacyl-tRNA synthetases hydrolyse ATP analogue, AMPPNP and to form ASN-AMP?
Are there any previous reports showing aaRSs exhibit hydrolysing activity towards AMP-PNP as in the case of HsAsnRS ?

"Following initial unsuccessful attempts to generate crystals in the presence of ATP and Asn, CDHsAsnRS was incubated in the presence of Asn and the ATP analogue, AMPPNP. Diffraction quality crystals were obtained; and we solved the structure (refined at 2.2 Å resolution), revealing the presence of Asn-AMP in the active site (Figure 4A,B; Supplementary Fig. 8)."

The interaction of E224 with the adenylate moiety of the ligand is not shown in Fig. 4B

Additional figure (superimposition of all three ligand bound structures) can be added to show a closer view of the important interactions between the bound ligands and the protein residues and flipping loop E279-F295 and Motif II Y321-E334 regions (as cartoon loop).

How the bound ion in the HsAsnRS/Ans-AMS structure is established as Na ion?
The coordination sphere/distance of bound ions (Mg and Na) in the HsAsnRS/Ans-AMP and HsAsnRS/Ans-AMS structures should be compared.

Page 10:

The word 'inset' needs to be fixed as 'insert'.

"In *P. falciparum*, the inset has a length of 76 amino acids."

Page 13:

The phrase may be replaced with 'a mouse model of *P. berghei* malaria.'
"ML901 exhibits excellent potency and selectivity, and effects single-dose cure in a mouse model of *P. falciparum* malaria."

Page 14:

The value and unit has to corrected
...extracts with 1 μ M OSM-S-106

The data shows that OSM-S-106 targets the charged tRNA, not inhibits tRNA charging, as the following sentence states.

"We showed that treatment of cultures with OSM-S-106 inhibits protein translation and triggers eIF2 α phosphorylation, which is diagnostic of the presence of uncharged tRNA^{16, 17}, providing further evidence that OSM-S-106 exerts its activity by inhibiting tRNA charging."

Page 23:

The X-ray source information is not provided for the data collection of HsAsnRS/AMP-Asn. Reference should be included for the HKL2000.

Crystals of ligand (Asn-OSM-S-106 and Asn-AMS) bound forms and empty form were obtained using purified apo CDHsAsnRS (native sequence) protein, but the whole N-terminal sequence information is not provided for 8TC7.

P11: The AMP-Asn bond was broken ..

P19: .. AMP-Asn in the initial phase of the aminoacylation reaction.

Figure 4:

In Fig. 4A, The ligand is not clearly visible and therefore the ligand may be drawn as space-filling model or molecular surface. The ligand Asn-AMP color should be matched with Fig. 4 B or C. In Fig. 4B and 4C, the view can be changed to show better clarity and interactions between protein residues may be avoided for clear view of protein-ligand interactions.

In Figure 4C, some residues are not labelled for panel (i).

In Figure 4C, glycerol (GOL) can be colored differently.

Figure 5:

Fig. 5B, the ligand molecule may be drawn as space-filling model or molecular surface. An additional panel should be added for the close-up view for bound ligand/tRNA and R487S. Ligand labels for panels F and G can be moved to the top panel.

Figure S1:

The full name for OPP can be added to the legend.

Figure S7:

The disordered loop residues can be marked either or both in Figure and legend.

Figure S8:

Ligand Asn-AMP color should be the same for all figures.

In Fig. S8A, The bound ligand is not shown for HsAsnRS'. The ligand is not clearly visible and therefore the ligand may be drawn as space-filling model or molecular surface.

In Fig. S8B, The loop residue ranges for the highlighted flipping loop and Motif II can be added. The difference Fourier/Omit maps for the bound ligand should be included.

Figure S9 and S10:

The above (Figure S8) comments apply to both of these figures

Table S3:

The abbreviation for SIM can be added as footnote

Table S8:

Several values should be corrected in the crystallographic table (Table S8) and many parameter values require decimal approximation. Some of the necessary changes are mentioned below

The title of each structure should clearly indicate the bound ligand

HsAsnRS/AMP-Asn should be HsAsnRS/Asn-AMP

HsAsnRS-AMS should be HsAsnRS/Ans-AMS

HsAsnRS-OSM106 should be HsAsnRS/OSM-S-106

X-ray 1 source can be added to this table

In the refinement section, the low resolution of the high resolution bin should be rounded to two decimal places as same as in the data collection section.

Rwork/Rfree should be three decimal approximation

The Willson and mean B factor should be rounded to a single decimal

The total number of Ligand/ion atoms is not correct for HsAsnRS-Asn-AMP (324 instead of 42)

Total number of non-H atoms for protein is not correct (13501 instead of 13783)

The hydrogen atoms of Asn-AMS seems to be accounted under the ligand/ion column (224 instead of 304) and therefore total number of non-hydrogen atoms is not match (protein+ligand/ion +solvent). This should be corrected.

For HsAsnRS-OSM106, the number of ligand/ion (346 instead of 410) and solvent (1031 instead of 1030) atoms to be corrected.

Reviewer #1

“...this paper has important implications for the entire field of drug discovery since it will be very important to test any sulfonamide drugs for its ability to form covalent adducts with various electrophiles in the cells, since the resulting conjugates can contribute to the observed pharmacological activity of sulfonamides. The paper is well written and can be accepted as is.”

We thank the Reviewer for their recognition of the significance of our work.

Reviewer #2

The manuscript by Xie et al. makes important contributions to the design of new therapeutics for Plasmodium.

We thank the Reviewer for their positive comment on our work.

A few important scholarly and technical concerns must be addressed before accepting the manuscript.

1. Can aminoacyl-tRNA synthetases hydrolyse AMPPNP?

Are there any studies reported on AMPPNP hydrolysis by aaRSs or other enzymes? The supporting references should be included.

We apologise that the text used for this section may have been confusing. AMPPNP is non-hydrolysable, *i.e.*, the beta-gamma phosphate bond is generally not susceptible to enzyme-catalysed hydrolysis. However, AMPPNP can serve as a substrate for some amino acyl tRNA synthetases (Freist et al., 1980) (Yang et al., 2006). The point of attack is the alpha-beta phosphate bond. To clarify this point, the text has been changed to “We employed the ATP analogue, AMPPNP, which can serve as a substrate for some aminoacyl tRNA synthetases (Freist et al., 1980) (Yang et al., 2006).” We have also included relevant references.

2. ATP may have been used and therefore an attempt should be made to model pyrophosphate (DPO) instead of imidodiphosphoric acid (2PN) and B-factor/occupancy can be compared.

A solution containing AMPPNP, Mg²⁺ and L-Asn was added directly to the apoHsAsnRS protein crystal. The structure of apoHsAsnRS was also solved and there was no evidence for co-purification of ATP. Therefore, we prefer not to model pyrophosphate, because ATP was neither present when the enzyme was purified nor added during crystallization steps.

3. The environment of the bound ion in the HsAsnRS/Asn-AMS structure indicating that this ion may be Mg²⁺ instead of Na ion. The coordination sphere/distance should be used to validate the bound ion.

The identity of the bound ion, coordination sphere, and distances were validated using the Check My Metal server. We have now included a statement in the Methods with the relevant reference (Gucwa et al., 2023).

4. The difference Fourier/Omit maps for the bound ligands should be included to verify unequivocally bound ligands. The above maps may be added for all four molecules (A-D) in the HsAsn/ASN-AMP structure.

We have generated omit maps for the bound ligands in each molecule of the Asymmetric Unit (ASU). Figure R1 below shows that the position of the ligand is well-supported by the electron density in all chains. The 2FoFc density maps in supplementary figures 8D, 9D, 10E have been replaced with representative omit maps.

Figure R1. Composite omit maps ($2mF_o$ - DF_c) for the bound ligands in the CDHAsnRS structures in complex with Asn-AMP, Asn-AMS and Asn-OSM-S-106. Maps contoured at 1.5σ .

5. Do all four molecules (A-D) in the ASU have the same conformation? particularly for active site residues, and Motif II in the different ligands bound structures.

This query is answered by the overlays in response to comment 6.

6. A structural overlay of molecules A-D for different ligand bound forms may be included. The active site residues & Motif II can be highlighted. This will allow to see if there are any noticeable or significant changes between the molecules.

Overlays of the ligands, active site residues and motifs in the ASU have been provided as additional panels in Supplementary Figures 8E, 9E and 10F.

7. The ligand is not clearly visible in the figures of cartoon representation (Fig. 4A, Figs. S8, S9 and S10) and therefore the ligand may be drawn as space-filling model or molecular surface. Several values should be corrected in the crystallographic table.

Fig. 4A is designed to illustrate the overall structure and relative position of the ligand. The emphasis is not on the ligand itself. We feel that a space filling models would obscure other important detail. In an effort to address the reviewer's request that the ligand be made more clearly visible, the representation has been modified with contrasting colours and thicker sticks.

8. There is no consistency in protein/ligand nomenclature. This should be maintained consistently throughout the manuscript (text, Figure legends and Tables). AMP-Asn has been used in some places instead of Ans-AMP. These needs to be fixed

All ligand names have been corrected for consistency.

Minor comments:

Asn-AMP/HsAsnRS (Page 12 and Figure S8 legend)

Asn-AMS/HsAsnRS (Figure S10 legend)

HsAsnRS/Asn-AMP

HsAsnRS/Asn-AMS

HsAsnRS/Asn-OSM-S-106 (Page 16 and Page 17)

Asn-OSM-S-106/HsAsnRS ((Figure S9 legend)

HsNRS/Asn-AMP (Figure 4)

AMP-Asn (Page 11 and Page 19)

Corrected.

Both Figure or Fig. notation have been used and these should be corrected.

Corrected.

"-" hyphen should be added to asparaginyl tRNA synthetase and aminoacyl tRNA synthetases

Corrected.

P. falciparum tyrosine RS (PfTyrRS) can be wrrien as P. falciparum tyrosine-tRNA synthetase (PfTyrRS).

Corrected.

delete "-" hyphen in Homo sapiens Tyr-RS (HsTyrRS)

Corrected.

The label notation of the mutant residue can be added to superscript PfAsnRSR487S instead of subscript PfAsnRSR487S

Corrected.

Page 4: Some IC50 values are with SD and some values are without SD

4.4 ± 2.6 μM

2.1 μM, 0.9 μM

Corrected.

Check the statement -- Conversion of the sulphonamide to a sulfamate (OSM-LO-80; Fig. 1E; 2.1 μM) or a sulfamate with an extended linker (OSM-LO-81; Fig. 1F; 0.9 μM) decreased activity (Table 1).

The activity is increased for OSM-LO-81 but statement is reporting that activity decreased. This needs to be fixed.

The sentence has been changed to "Conversion of the sulfonamide to either a sulfamate (OSM-LO-80; Fig. 1E; 2.1 ± 0.4 μM) or a sulfamate with an extended linker (OSM-LO-81; Fig. 1F; 0.93 ± 0.22 μM) decreased inhibitory activity compared with OSM-S-106 (Table 1)."

Page 5:

Abbreviations for SNV and CNV can be added where they first appear

Corrected.

IC90 values should be corrected

3 x IC90 (507.7 nM) Page 5

3 x IC90 (508 nM) Page 18

Text for both mentions is now "3 x IC90 (508 nM)"

The abbreviation of dhodh should be mentioned.

Corrected.

More clarity is required in the following sentence.

"This yielded 14 recrudescant wells (corresponding to a Minimum Inoculum for Resistance (MIR) of 1.4 ×

_106), with evidence of separate amplification events encompassing the dhodh locus consistent with increased IC50 values (Supplementary Table 3,4; Supplementary Dataset 2).”

The sentence has been changed to: “This yielded 14 recrudescence wells, which corresponds to a Minimum Inoculum for Resistance (MIR) of 1.4×10^6 parasites required to obtain resistance. Whole-genome sequence analysis provided evidence of amplification events encompassing the dhodh locus, consistent with increased IC₅₀ values for DSM265 (Supplementary Table 3,4; Supplementary Dataset 2).”

“This selection yielded 21 newly emerged coding variants in 10 unique core genes.

The sentence is changed to: This ramp-up selection yielded 21 newly emerged coding variants in 10 unique core genes.

Page 6:

“-” hyphen should be deleted with a blank space - 6-h exposure

Corrected.

The prolyl-tRNA synthetase inhibitor halofuginone also induces eIF2 α phosphorylation. This can be mentioned with reference

A reference has been added - doi: 10.1021/acsinfecdis.8b00363. The Bridgford et al reference has also been replaced, as that study did not look at aaRS inhibitors.

Page 8:

Mujumdar et al., 2018 is not in the reference list. The number system should be used for this reference invoking

Corrected.

Page 9 and Page 10:

Can aminoacyl-tRNA synthetases hydrolyse ATP analogue, AMPPNP and to form ASN-AMP?

Are there any previous reports showing aaRSs exhibit hydrolysing activity towards AMP-PNP as in the case of HsAsnRS ?

“Following initial unsuccessful attempts to generate crystals in the presence of ATP and Asn, CDHsAsnRS was incubated in the presence of Asn and the ATP analogue, AMPPNP. Diffraction quality crystals were obtained; and we solved the structure (refined at 2.2 Å resolution), revealing the presence of Asn-AMP in the active site (Figure 4A,B; Supplementary Fig. 8).”

As described above, we have included a reference to explain the production of ASN-AMP from AMPPNP and Asn.

The interaction of E224 with the adenylate moiety of the ligand is not shown in Fig. 4B

The interacting residue should be E324 instead of E224. We have made the correction in the main text. The interaction of E324 with the adenylate moiety has now been added to Figure 4B, Supp Figure 8C and Supp Figure 9C.

Additional figure (superimposition of all three ligand bound structures) can be added to show a closer view of the important interactions between the bound ligands and the protein residues and flipping loop E279-F295 and Motif II Y321-E334 regions (as cartoon loop).

We have now made an additional Supp Figure (Supp Figure S11) to show the superimposition of all three ligand-bound structures.

How the bound ion in the HsAnsRS/Ans-AMS structure is established as Na ion?. The coordination sphere/distance of bound ions (Mg and Na) in the HsAnsRS/Ans-AMP and HsAnsRS/Ans-AMS structures should be compared.

Please see the response to Comments 3. The coordination sphere and distances were validated using Check My Metal.

Page 10:

The word 'inset' needs to be fixed as 'insert'. "In P. falciparum, the inset has a length of 76 amino acids."
Corrected.

Page 13:

The phrase may be replaced with "a mouse model of P. berghei malaria."

"ML901 exhibits excellent potency and selectivity, and effects single-dose cure in a mouse model of P. falciparum malaria."

This mouse that is used for these studies is the humanized NOD-scid IL2Rnull mouse model infected with *P. falciparum*. To make this clear, we have changed the phrase to "... a humanised mouse model of *P. falciparum* malaria".

Page 14:

The value and unit has to corrected ...extracts with 1 mM OSM-S-106.

Corrected.

The data shows that OSM-S-106 targets the charged tRNA, not inhibits tRNA charging, as the following sentence states.

"We showed that treatment of cultures with OSM-S-106 inhibits protein translation and triggers eIF2 α phosphorylation, which is diagnostic of the presence of uncharged tRNA^{16, 17}, providing further evidence that OSM-S-106 exerts its activity by inhibiting tRNA charging."

The sentence has been changed to "OSM-S-106 activity leads to a decrease in the level of charged tRNA"

Page 23:

The X-ray source information is not provided for the data collection of HsAsnRS/AMP-Asn.

The X-ray source is the Korean Synchrotron: PAL/PLS BEAMLIN 5C. This information has now been included in the Methods section.

Reference should be included for the HKL2000.

This information has now been included in the Methods section (Otwinowski and Minor, 1997).

Crystals of ligand (Asn-OSM-S-106 and Asn-AMS) bound forms and empty form were obtained using purified apo CDHsAsnRS (native sequence) protein, but the whole N-terminal sequence information is not provided for 8TC7.

The sequence for CDHsAsnRS was described in the Biological Method section. We have now added the sequence info "residues A98–P548" to the main text.

P11: The AMP-Asn bond was broken ..

P19: .. AMP-Asn in the initial phase of the aminoacylation reaction.

Standardised to Asn-AMP.

Figure 4:

In Fig. 4A, The ligand is not clearly visible and therefore the ligand may be drawn as space-filling model or molecular surface. The ligand Asn-AMP color should be matched with Fig. 4 B or C.

As detailed above, we feel that space filling models would obscure other important detail. In an effort to address the reviewer's request that the ligand be made more clearly visible, the representation has been modified with contrasting colours and thicker sticks. The Asn-AMP ligand colour has been made uniform across all figures.

In Fig. 4B and 4C, the view can be changed to show better clarity and interactions between protein residues may be avoided for clear view of protein-ligand interactions.

We have already included 2 orientations for each structure and the models are available on the PDB. Readers can inspect the PDB structures for more detail. We feel it is not necessary to add additional views, but would be willing to do so if the Editor thinks this is necessary.

In Figure 4C, some residues are not labelled for panel (i).

We have now included all relevant labels.

In Figure 4C, glycerol (GOL) can be colored differently.

We feel that this would not assist with the interpretation of the data as glycerol (GOL) is not the ligand of interest, but would be willing to do so if the Editor thinks this is necessary.

Figure 5:

Fig. 5B, the ligand molecule may be drawn as space-filling model or molecular surface.

As detailed above, we feel that a space filling models would obscure other important detail. In an effort to address the reviewer's request that the ligand be made more clearly visible, the representation has been modified with contrasting colours and thicker sticks.

An additional panel should be added for the close-up view for bound ligand/tRNA and R487S.

We have now rearranged the panels in Figure 5A to adopt a vertical layout, and added an additional panel (close-up view) as Figure 5Bii.

Ligand labels for panels F and G can be moved to the top panel.

We have reorganised this figure and all the ligand labels are positioned consistently in each panel.

Figure S1: The full name for OPP can be added to the legend.

Corrected.

Figure S7: The disordered loop residues can be marked either or both in Figure and legend.

We have now added text to the figure legend, indicating the disordered loop residues.

“(B) Ribbon diagram of the active site of apo HsAsnRS showing the positions of the flipping loop (E279 - T283) and motif II loop (Y321 – E334; yellow).”

Figure S8: Ligand Asn-AMP color should be the same for all figures, including LigPlots.

The Asn-AMP ligand colour has been made uniform across all figures.

In Fig. S8A, The bound ligand is not shown for HsAsnRS'. The ligand is not clearly visible and therefore the ligand may be drawn as space-filling model or molecular surface.

As discussed above the representation has been modified with contrasting colours and thicker sticks. We have also added the ligand to the other monomer.

In Fig. S8B, The loop residue ranges for the highlighted flipping loop and Motif II can be added.

The loop residue ranges for the highlighted flipping and Motif II loops have been added to the figure legend.

The difference Fourier/Omit maps for the bound ligand should be included.

Please see response to Comment 4.

Figure S9 and S10:

The above (Figure S8) comments apply to both of these figures

Please see response to Comment 4.

Table S3:

The abbreviation for SIM can be added as footnote

Corrected.

Table S8:

Several values should be corrected in the crystallographic table (Table S8) and many parameter values require decimal approximation. Some of the necessary changes are mentioned below

The title of each structure should clearly indicate the bound ligand

HsAsnRS/AMP-Asn should be HsAsnRS/Asn-AMP

HsAsnRS-AMS should be HsAsnRS/Asn-AMS

HsAsnRS-OSM106 should be HsAsnRS/OSM-S-106

Corrected.

X-ray 1 source can be added to this table

The X-ray source is the Korean Synchrotron: PAL/PLS BEAMLIN 5C. This information has now been included in the Table.

In the refinement section, the low resolution of the high resolution bin should be rounded to two decimal places as same as in the data collection section.

Corrected.

Rwork/Rfree should be three decimal approximation. The Willson and mean B factor should be rounded to a single decimal.

Corrected.

The total number of Ligand/ion atoms is not correct for HsAsnRS-Asn-AMP (324 instead of 42)

Total number of non-H atoms for protein is not correct (13501 instead of 13783)

Corrected.

The hydrogen atoms of Asn-AMS seems to be accounted under the ligand/ion column (224 instead of 304) and therefore total number of non-hydrogen atoms is not match (protein+ligand/ion +solvent). This should be corrected.

Corrected.

For HsAsnRS-OSM106, the number of ligand/ion (346 instead of 410) and solvent (1031 instead of 1030) atoms to be corrected.

Corrected.

We thank the Reviewers for their constructive comments and hope you will now find the manuscript suitable for publication in Nature Communications.

Regards

Leann Tilley

Response References

Freist, W., H. Wiedner, and F. Cramer. 1980. Chemically modified ATP derivatives for the study of aminoacyl-tRNA synthetases from baker's yeast: ATP analogs with fixed conformations or modified triphosphate chains in the aminoacylation reaction. *Bioorganic Chemistry*. 9:491-504.

- Gucwa, M., J. Lenkiewicz, H. Zheng, M. Cymborowski, D.R. Cooper, et al. 2023. CMM-An enhanced platform for interactive validation of metal binding sites. *Protein Sci.* 32:e4525.
- Otwinowski, Z., and W. Minor. 1997. Processing of X-ray diffraction data collected in oscillation mode. *Methods Enzymol.* 276:307-326.
- Yang, X.-L., F.J. Otero, K.L. Ewalt, J. Liu, M.A. Swairjo, et al. 2006. Two conformations of a crystalline human tRNA synthetase–tRNA complex: implications for protein synthesis. *Embo J.* 25:2919-2929.

REVIEWERS' COMMENTS

Reviewer #2 (Remarks to the Author):

The authors have implemented all the points I raised in the revised version, the manuscript can be accepted for publication.